# Skeletal muscle-secreted DLPC orchestrates systemic energy homeostasis by enhancing adipose browning

Xiaodi Hu[1,9], Mingwei Sun[2,9], Qian Chen[1], Yixia Zhao[1], Na Liang[1], Siyuan Wang[3], Pengbin Yin [4,5], Yuanping Yang[2], Sin Man Lam [6,7], Qianying Zhang[1], Alimujiang Tudiyusufu[1], Yingying Gu[1], Xin Wan [1], Meihong Chen[1], Hu Li[2], Xiaofei Zhang [2], Guanghou Shui [6,7], Suneng Fu[8], Licheng Zhang[4,5], Peifu Tang[4,5], Catherine C. L. Wong [3], Yong Zhang [1,2] ✉ & Dahai Zhu [1,2] ✉

MyoD is a skeletal muscle-specifically expressed transcription factor and plays a critical role in regulating myogenesis during muscle development and regeneration. However, whether myofibers-expressed MyoD exerts its metabolic function in regulating whole body energy homeostasis in vivo remains largely unknown. Here, we report that genetic deletion of *Myod* in male mice enhances the oxidative metabolism of muscle and, intriguingly, renders the male mice resistant to high fat diet-induced obesity. By performing lipidomic analysis in muscle-conditioned medium and serum, we identify 1,2-dilinoleoyl-sn-glycero-3-phosphocholine (DLPC) as a muscle-released lipid that is responsible for MyoD-orchestrated body energy homeostasis in male *Myod* KO mice. Functionally, the administration of DLPC significantly ameliorates HFD-induced obesity in male mice. Mechanistically, DLPC is found to induce white adipose browning via lipid peroxidation-mediated p38 signaling in male mice. Collectively, our findings not only uncover a novel function of MyoD in controlling systemic energy homeostasis through the muscle-derived lipokine DLPC but also suggest that the DLPC might have clinical potential for treating obesity in humans.

Obesity is increasing in prevalence worldwide and represents a major health burden because of its metabolic complications, which include diabetes and cardiovascular disease[1–3]. It has been estimated that about 58% of adults in the world will be overweight or will have obesity by the year 2030[4]. Therefore, obesity has been identified as a major emerging chronic disease of the 21st century[5]. Whole-body metabolic homeostasis is orchestrated via the collaborative effort of various metabolic tissues, including liver, skeletal muscle, and adipose tissue[6], which communicate by secreting circulating factors such as IL-6[7] and irisin[8]. Dysregulation of inter-tissue communication can lead to the

[1]State Key Laboratory for Complex, Severe, and Rare Diseases, Institute of Basic Medical Sciences, Chinese Academy of Medical Sciences and School of Basic Medicine, Peking Union Medical College, Beijing 100005, China. [2]Bioland Laboratory (Guangzhou Regenerative Medicine and Health Guangdong Laboratory), Guangzhou 510005, China. [3]Clinical Research Institute, State Key Laboratory for Complex, Severe, and Rare Diseases, Peking Union Medical College Hospital, Chinese Academy of Medical Science & Peking Union Medical College, Beijing 100730, China. [4]Senior Department of Orthopedics, The Fourth Medical Center of Chinese PLA General Hospital, Beijing 100853, China. [5]National Clinical Research Center for Orthopedics, Sports Medicine & Rehabilitation, Beijing 100853, China. [6]LipidALL Technologies Company Limited, Changzhou 213022, China. [7]State Key Laboratory of Molecular Developmental Biology, Institute of Genetics and Developmental Biology, Chinese Academy of Sciences, Beijing 100101, China. [8]Guangzhou Laboratory, Guangzhou 510005, China. [9]These authors contributed equally: Xiaodi Hu, Mingwei Sun. ✉e-mail: yongzhang@ibms.pumc.edu.cn; dhzhu@pumc.edu.cn

onset of obesity and diabetes[9]. Adipose tissue is well known to be functionally significant in mediating whole-body energy homeostasis and obesity development[10]. Interestingly, recent studies have demonstrated that lipokines, which are lipid species predominantly secreted from adipose tissue, play critical roles in controlling whole-body energy homeostasis by communicating with other metabolic organs, such as liver and muscle[11–14].

Skeletal muscle is the largest endocrine tissue. It secretes peptides that regulate inter-tissue communication to mediate the whole-body energy balance[15]. Skeletal muscle is also a highly metabolically active tissue; it contains many intermediate metabolites that are important for the energy production needed to sustain skeletal muscle function during development and under exercise and stress[7,16–19]. An interesting question is whether skeletal muscle, like adipose tissue, also secretes lipokines that act as metabolic regulators in controlling whole-body energy homeostasis via crosstalking with fat and/or liver. An emerging focus is to identify a regulatory lipokine secreted from skeletal muscle and then elucidate its underlying molecular mechanism in orchestrating inter-organ crosstalk to mediate whole-body metabolic homeostasis and/or the development of obesity and diabetes[20].

MyoD is a well-known master transcription factor (TF) for myogenic cell lineage determination and differentiation during skeletal muscle development and regeneration[21–25]. Loss of MyoD induces cell fate transdifferentiation of myoblasts into brown adipocytes[26]. To date, most of the research on MyoD has focused on its function as a TF in regulating the transcription of myogenic genes in myoblasts[21,27,28]. However, non-canonical roles of MyoD were recently reported: our group discovered that MyoD functions as a "genome organizer" by specifying 3D genome architecture unique to muscle cell development[29]. In addition, a few studies have suggested that myofiber-expressed MyoD may mediate adult skeletal muscle metabolism[30–35]. In skeletal muscle, MyoD acts via an alternative NF-κB signaling pathway to regulate transcription of the metabolic gene, *PGC-1β*, which encodes a master regulator of mitochondrial biogenesis[34]. We recently showed that myofiber-expressed MyoD acts as a metabolic sensor to regulate transcription of *csf3*; this gene encodes G-CSF protein, which was identified as a myofiber-released metabolic niche factor required for the establishment and maintenance of Pax7[Hi] muscle stem cells in mice[32]. Molecularly, MyoD ChIP-seq analysis using adult skeletal muscle found that MyoD exhibits binding peaks on numerous metabolic genes[35].

In this study, we used *Myod* knockout mice (KO) as an in vivo system to directly investigate the function of MyoD in regulating the muscle metabolism of adult mice. We show that MyoD does, indeed, critically regulate the expression of genes encoding fatty acid oxidation-related enzymes in mice. We further reveal that MyoD has a systemic function in mediating global metabolism by showing that *Myod* KO mice are resistant to high-fat diet (HFD)-induced obesity. These observations indicate that muscle-expressed MyoD might control whole-body metabolic homeostasis by communicating with adipose tissue via secretion of signaling lipids, and that *Myod* KO mice might provide a genetic system for identifying muscle-secreted lipids in mice.

In sum, by performing lipidomic analysis in muscle-conditioned medium and sera from WT and *Myod* KO mice, we herein uncover 1,2-dilinoleoyl-sn-glycero-3-phosphocholine (DLPC) as a muscle-released lipid that is functionally responsible for MyoD-orchestrated body energy homeostasis by inducing inguinal white-adipose tissue (iWAT) browning in mice. Mechanistically, we show that DLPC induces iWAT browning via lipid peroxidation-mediated p38 MAPK activation both in vitro and in mice. Based on our collective findings, we herein propose DLPC as the first reported lipokine secreted by skeletal muscle and suggest that it could have clinical potential for treating obesity in humans.

## Results

### *Myod* is upregulated in TA muscle, but not in Sol muscle, in response to HFD feeding

Skeletal muscle is known to play important roles in maintaining whole-body metabolic homeostasis[16], but the underlying molecular mechanisms remain largely unknown. In this study, we aimed to elucidate the relevant molecular mechanism by identifying key factors that are specifically expressed in skeletal muscles and function to regulate whole-body metabolic homeostasis via crosstalk between skeletal muscle and fat in mice fed with HFD. To this end, we first performed RNA-seq analysis using glycolytic *tibialis anterior* (TA) and oxidative *soleus* (Sol) muscles isolated from 8-week-old C57BL/6j male mice that had been fed HFD or standard diet (SD) for 2 weeks (Fig. 1a). Our transcriptomics profiling demonstrated that both TA and Sol muscles could actively respond to HFD feeding, as evidenced by the identification of differentially expressed genes (DEGs) between the HFD and SD groups (Fig. 1b). TA muscle yielded more DEGs than Sol muscle: we identified 124 up- and 302 down-regulated genes in TA, compared to 96 and 202, respectively, in Sol (cut off: adjusted $p < 0.05$, fold-change (FC) > 1.2) (Supplementary Fig. 1a). Enriched gene ontology (GO) analysis of the DEGs revealed that Sol and TA muscles showed differential responses to HFD feeding. In Sol muscle, the upregulated genes were enriched for GO terms related to fatty acid oxidation and the downregulated genes were enriched for those related to lipid storage (Fig. 1c). This suggested that Sol muscle may contribute to regulating lipid oxidation in mice. The genes upregulated in TA muscle, in contrast, were unrelated to lipid oxidation (Fig. 1d). This indicated that the glycolytic TA muscle has a weak capability to consume fatty acids. Our recent metabolic flux analysis also supported the notion that the TA muscle has relatively low β-oxidation activity[36].

Next, we searched for the transcription factors (TFs) that might regulate the differential responses of TA and Sol muscles to HFD (Fig. 1e and Supplementary Fig. 1b). We were particularly interested in identifying TFs that were specifically expressed in muscle and differentially expressed between TA and Sol muscle in response to HFD. Using these criteria, we identified a single TF: Muscle-specific *Myod* was found to be significantly upregulated in TA muscle but not in Sol muscle in response to HFD in mice (Fig. 1e, f). Our group and others previously reported that MyoD is predominantly expressed in glycolytic muscle and functionally regulates glycolytic metabolism[30,32]. Thus, the upregulation of *Myod* in glycolytic TA muscle in response to HFD might be a negative effect on oxidative metabolism to consume excess fatty acids.

### *Myod* KO mice exhibit enhanced oxidative metabolism in skeletal muscle

Next, we explored the potential function of MyoD in mediating muscle metabolism by examining the expression of metabolic enzyme-encoding genes[37–42] in the skeletal muscle of *Myod* knockout (KO) mice. We observed significant upregulation of fatty acid catabolism and mitochondrial biogenesis-related genes in glycolytic *quadriceps* (Qu) muscle of *Myod* KO mice compared to WT littermates; the altered genes included those encoding long-chain acyl-CoA synthetase (*ACSL*), fatty acid binding protein 3 (*FABP3*), peroxisome proliferator-activated receptor gamma coactivator 1 beta (*PGC-1β*), peroxisome proliferator-activated receptor alpha (*PPARα*), and cytochrome c (*Cytc*) (Supplementary Fig. 2a, b). In the same muscle, we observed *Myod* KO-related downregulation of genes encoding key enzymes for the glycolysis pathway, including hexokinase 2 (*HK2*), phosphofructokinase in muscle (*PFKm*), and lactate dehydrogenase A (*LDHA*) (Supplementary Fig. 2c). A function of MyoD in regulating glycolytic metabolism in skeletal muscle was further supported by our observation that oxidative type I and type IIa fibers were increased and glycolytic type IIb fibers were decreased in *Myod* KO mice (Supplementary Fig. 2d).

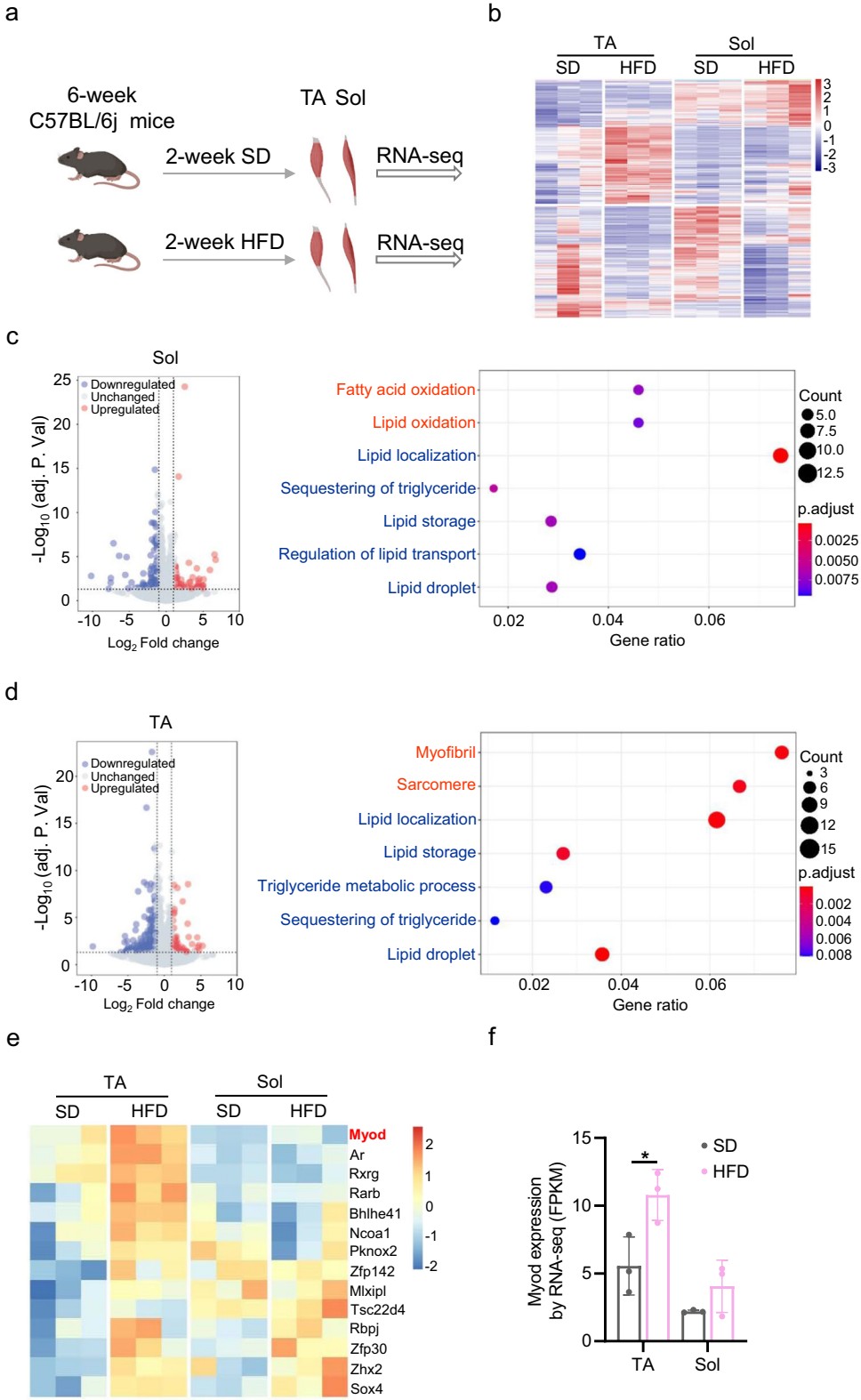

Since enhanced oxidative metabolism in skeletal muscle generally benefits systemic energy homeostasis[16], we reasoned that the elevated oxidative metabolism in *Myod* KO muscle might contribute to differences in whole-body metabolic homeostasis. To directly test this possibility, we used metabolic chambers to determine the effects of *Myod* deletion on whole-body energy parameters under a normal chow feeding state. We found that *Myod* KO mice showed significantly

elevated oxygen consumption (Supplementary Fig. 2e), carbon dioxide production (Supplementary Fig. 2f), and energy expenditure (Supplementary Fig. 2g), indicating that the basic metabolic rate was higher in *Myod* KO mice than in WT littermates. Of note, the ambulatory activity did not differ between the KO and WT littermates (Supplementary Fig. 2h), thus, the enhanced oxidative metabolism in *Myod* KO mice was not simply due to increased movement. Moreover, we

**Fig. 1 | *Myod* is upregulated in TA muscle, but not in Sol muscle, in response to HFD feeding. a** Experimental scheme for RNA-seq analysis of *tibialis anterior* (TA) and *soleus* (Sol) muscles from 8-week-old C57BL/6j male mice that had been fed with high-fat diet (HFD) or standard diet (SD) for 2 weeks, *n* = 3 mice. Graphs in (**a**) is generated by BioRender (https://www.biorender.com). **b** Heatmap showing the differentially expressed genes (DEGs) between the HFD and SD control mice described in (**a**), as determined by RNA-seq, *n* = 3 mice. **c** Volcano plot (left) showing DEGs in Sol muscle and bubble plot (right) showing gene ontology (GO) terms enriched among the upregulated (red) and downregulated (blue) genes. **d** Volcano plot (left) showing DEGs in TA muscle and bubble plot (right) showing GO terms enriched among the upregulated (red) and downregulated (blue) genes. **e** Heatmap showing the transcription factors (TFs) upregulated in TA and Sol muscles from HFD-fed mice compared to SD controls. **f** *Myod* mRNA levels in TA and Sol muscles from the HFD-fed mice and SD controls described in (A), determined by RNA-seq, *n* = 3 mice. *p* = 0.0116 (SD-TA vs. HFD-TA). Data are presented as mean ± SD. Significance was assessed by two-way ANOVA (**f**). *$p < 0.05$ compared to SD control group. Differential genes represented in the volcano plot (**c**, **d**) were identified with a false discovery rate (FDR) < 0.05 (*p*-values adjusted using the Benjamini-Hochberg method) and a fold change >1.2, employing DESeq2. The enrichment analysis of Gene Ontology (GO) terms among the differentially expressed genes was performed using ClusterProfiler. A significance cutoff of *p*.adjust < 0.05 and *q* value < 0.05 was applied as the criteria for determining enriched terms (**c**, **d**). Source data are provided as a Source Data file.

measured core temperature of *Myod* KO and their WT controls. The core temperature in *Myod* KO mice were significantly higher than that in WT controls at either thermal neutral or cold conditions (Supplementary Fig. 3a, b), which is consistent with metabolic chamber analysis showing higher heat production or energy expenditure in *Myod* KO mice than in WT controls. To further evaluate the effect of *Myod* in heat production/energy expenditure, we performed mitochondrial respiration assay in the differentiated myotubes. To this end, the primary myoblasts isolated from the *Myod*^f/f mice were induced to differentiation for 48 h and then infected with adenovirus expressing Cre (Ad-Cre) to achieve deletion of *Myod* (cKO), infection with the adenovirus expressing EGFP (Ad-EGFP) as controls (cWT) (Supplementary Fig. 3c). This approach achieved successful knockout of *Myod* in the cKO myotubes, as measured by RT-qPCR (Supplementary Fig. 3d) and the cKO myotubes were as healthy as cWT controls (Supplementary Fig. 3e). Consistently, we found that the cKO myotubes exhibited higher oxygen consumption rate (OCR) than that in cWT controls (Supplementary Fig. 3f). Taken together, our data suggest that muscle-expressed MyoD might function as a metabolic regulator to help orchestrate the systemic energy balance under physiological conditions and/or HFD feeding in mice.

### *Myod* KO mice resist HFD-induced obesity

To functionally confirm that MyoD is a metabolic mediator of whole-body energy homeostasis in vivo, we examined whether deletion of *Myod* could have protective effects on HFD-induced obesity. We fed *Myod* KO mice and WT littermates with SD or HFD for 12 weeks and measured body weight, food intake, fat mass, and energy expenditure. We found that *Myod* KO mice had a significantly lower weight gain than WT littermates after HFD feeding (Fig. 2a), even though the KO mice ate slightly more food (Supplementary Fig. 2i). HFD-fed *Myod* KO mice also had significantly less fat mass than WT littermates (Fig. 2b, c) and significantly smaller size of adipocytes in inguinal white-adipose tissue (iWAT) and gonadal WAT (gWAT) (Fig. 2d). These findings indicate that *Myod* KO protects mice against HFD-induced obesity.

To further corroborate the function of MyoD as a metabolic regulator in controlling organismal energy homeostasis in mice, we used the metabolic chamber to measure the whole-body energy metabolism of *Myod* KO mice fed with HFD. We found that *Myod* KO mice fed with HFD had elevated oxygen consumption (Fig. 2e), carbon dioxide production (Fig. 2f) and energy expenditure (Fig. 2g), which nicely supported their lean phenotype. Consistently, *Myod* KO mice showed almost normal results on the GTT (glucose tolerance test) and ITT (insulin tolerance test) after 12 weeks of HFD feeding (Fig. 2h, i, Supplementary Fig. 2j, k), while WT littermates suffered impaired glucose tolerance and decreased insulin sensitivity (Fig. 2h, i, Supplementary Fig. 2j, k). We also observed lower level of triglycerides (TG) and total cholesterol (T-CHO) in sera of HFD-fed *Myod* KO mice compared to WT littermates (Fig. 2j, k). Collectively, our results provide the functional evidence that muscle-expressed MyoD is a bona fide metabolic mediator that plays an unanticipated regulatory role in controlling whole-body energy homeostasis in vivo.

### iWAT browning occurs in *Myod* KO mice

Next, we systematically analyzed the cellular phenotype of adipose tissues in *Myod* KO mice and WT controls. We first examined adipocyte numbers by measuring the genomic DNA contents in individual iWAT fat pads. There was no difference in the DNA contents of iWAT between KO mice and WT littermates (Supplementary Fig. 4a), indicating that the deletion of *Myod* did not affect the adipocyte number. These data indicate that adipogenesis do not significantly contribute to the reduced fat deposition in *Myod* KO mice fed with HFD. We then measured lipolysis by testing the expression levels of genes encoding the lipolysis-related enzymes, hormone-sensitive lipase (*HSL*) and adipose triglyceride lipase (*ATGL*). There was no overt difference in the expression levels of these genes between the two groups (Supplementary Fig. 4b), indicating HSL and ATGL are not transcriptionally regulated in adipose tissues of *Myod* KO mice.

Notably, histological analysis of iWAT demonstrated that uncoupling protein 1 (Ucp1), which is a molecular marker for iWAT browning, was significantly induced in *Myod* KO mice fed with SD or HFD (Fig. 3a). The increased level of Ucp1 were further confirmed by Western blot (Fig. 3b) and RT-qPCR (Fig. 3c, d) analyses of iWAT tissues obtained from SD- and HFD-fed mice. These data indicate that *Myod* KO mice exhibit increased iWAT browning. Further corroborating this, we detected elevated expression of two additional molecular markers of iWAT browning: the genes encoding elongation of very long chain fatty acids protein 3 (*Elovl3*) and cell death-inducing DFFA (DNA fragmentation factor-alpha)-like effector A (*Cidea*) (Fig. 3c, d). iWAT browning is usually concomitant with increased mitochondrial number, elevated mitochondrial respiration, and enhanced fatty acid oxidation. We measured the mitochondrial number in iWAT by quantifying the mitochondrial DNA content. As shown in Fig. 3e, the mitochondrial DNA content was significantly higher in *Myod* KO mice compared to WT littermates, regardless of diet, indicating that the mitochondrial number in iWAT is greater in *Myod* KO mice than in WT littermates. We also observed upregulation of the mitochondria biogenesis-related genes, mitochondrial transcription factor A (*Tfam*) and nuclear respiratory factor 1 (*Nrf1*) (Fig. 3f). Together, these results reveal that genetic deletion of skeletal muscle-expressed *Myod* resulted in iWAT browning, which might contribute to the reduced fat deposition and resistance to HFD-induced obesity seen in *Myod* KO mice.

### DLPC induces iWAT browning in HFD-fed *Myod* KO mice

Given our findings that deletion of *Myod* increases lipid and fatty acid oxidation in skeletal muscle in response to HFD feeding in mice and iWAT browning occurs in the *Myod* KO mice, it is conceivable that metabolites secreted by skeletal muscle might act as signaling factors to trigger iWAT browning in *Myod* KO mice via muscle-fat crosstalk. To examine this possibility, *Myod* KO mice and WT littermates were fed with HFD for 2 weeks, the collected TA and Sol muscles were subjected to RNA-seq, and Qu muscle was used to prepare muscle-conditioned medium (CM) for metabolomics and lipidomics analyses (Fig. 4a). Sera collected from the mice were also subjected to metabolomics and lipidomics profiling (Fig. 4a). Our transcriptome

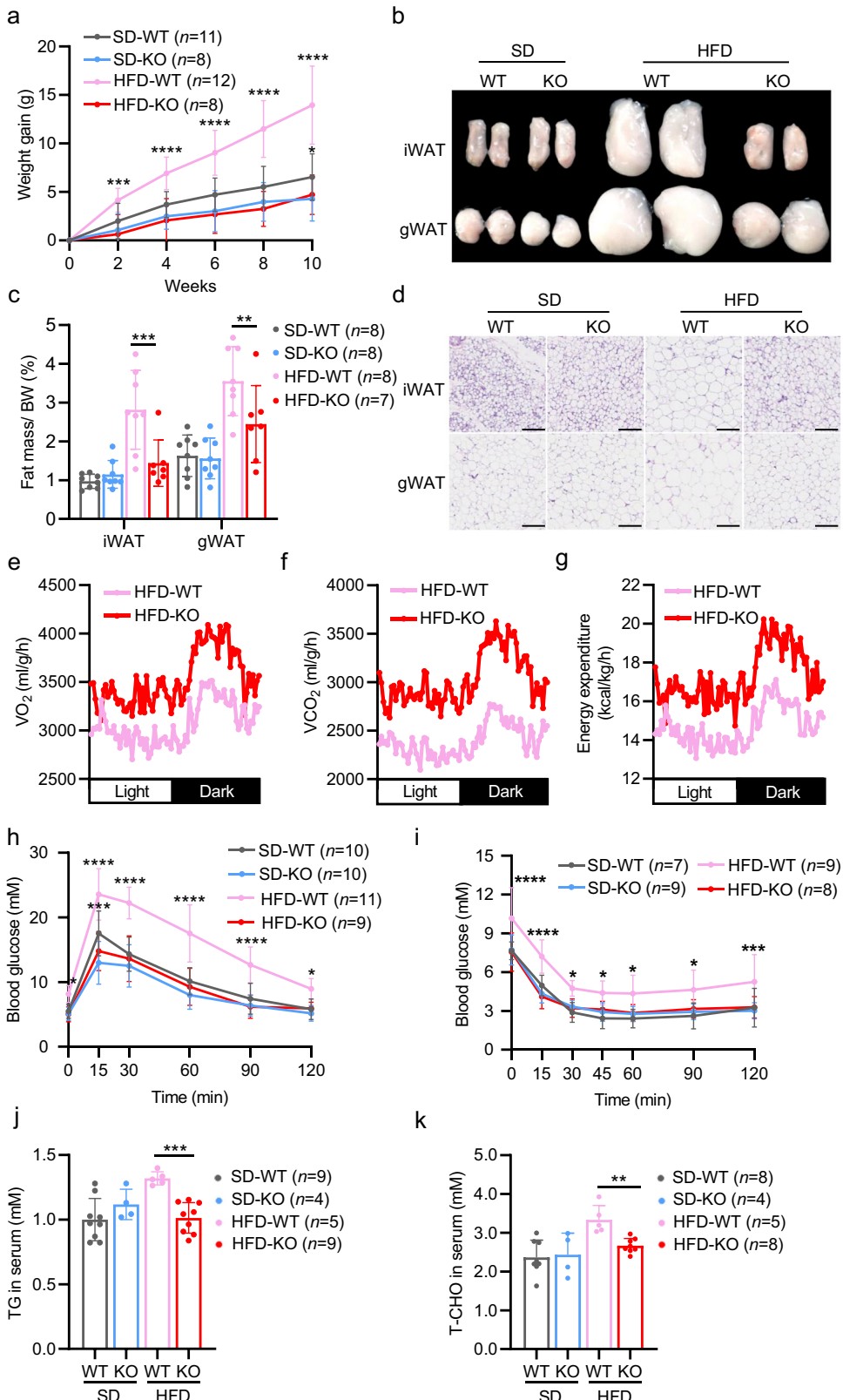

profiling demonstrated that the DEGs between *Myod* KO and WT littermates were enriched in GO terms for fatty acid catabolism and lipid oxidation (Fig. 4b, Supplementary Fig. 5a), which provides further support for our hypothesis that muscle-secreted metabolites or lipids may mediate iWAT browning in *Myod* KO mice. However, the untargeted metabolomics analyses of muscle-derived CM and sera showed no overt difference between *Myod* KO mice and WT littermates

(Supplementary Data. 1, 2). Notably, lipidomics analysis demonstrated that phosphatidylcholine (PC) was significantly elevated in the muscle-derived CM (Fig. 4c, Supplementary Data. 3) and sera (Fig. 4d, Supplementary Data. 4) of *Myod* KO mice compared to WT littermates. Measurement of the PC concentration in skeletal muscle (Qu) using a chemical kit confirmed that the PC levels were higher in the skeletal muscle of *Myod* KO mice compared to WT littermates (Fig. 4e). We

**Fig. 2 | *Myod* KO mice exhibit enhanced oxidative metabolism in skeletal muscle and resist HFD-induced obesity. a** Weight gain of *Myod* KO mice and WT littermates fed with SD or HFD for 10 weeks. For SD-WT vs. SD-KO, *p* = 0.0443 (10-week); For HFD-WT vs. HFD-KO, *p* = 0.0005 (2-week), *p* < 0.0001 (4-week), *p* < 0.0001 (6-week), *p* < 0.0001 (8-week), *p* < 0.0001 (10-week). **b** Fat pads from the mice described in (**a**). iWAT, inguinal white-adipose tissue. gWAT, gonadal white-adipose tissue. **c** Fat masses from the mice described in (**a**), For iWAT, *p* = 0.001 (HFD-WT vs. HFD-KO); For gWAT, *p* = 0.009 (HFD-WT vs. HFD-KO). **d** Representative hematoxylin and eosin (H&E)-stained sections of iWAT and gWAT from the mice described in (**a**). Scale bar, 100 μm. **e–g** O$_2$ consumption (**e**), CO$_2$ production (**f**) and energy expenditure (**g**) by *Myod* KO mice and WT littermates fed with HFD, as determined by metabolic chamber analysis, *n* = 4 mice. **h** Glucose

tolerance test (GTT) results of the mice described in (**a**), For SD-WT vs. SD-KO, *p* = 0.0003 (15 min); For HFD-WT vs. HFD-KO, *p* = 0.0192 (0 min), *p* < 0.0001 (15 min), *p* < 0.0001 (30 min), *p* < 0.0001 (60 min), *p* < 0.0001 (90 min), *p* = 0.0151 (120 min). **i** Insulin tolerance test (ITT) results of the mice described in (**a**). For HFD-WT vs. HFD-KO, *p* < 0.0001 (0 min), *p* < 0.0001 (15 min), *p* < 0.0136 (30 min), *p* = 0.0450 (45 min) *p* = 0.0146 (60 min), *p* = 0.0166 (90 min), *p* = 0.0009 (120 min). **j** Triglyceride (TG) levels in sera of the mice described in (**a**), *p* = 0.0002 (HFD-WT vs. HFD-KO). **k** Total cholesterol (T-CHO) levels in sera of the mice described in (**a**), *p* = 0.0011 (HFD-WT vs. HFD-KO). Data are presented as mean ± SD. Significance was assessed by two-way ANOVA (**a, c, h, i**) or two tail Student's *t*-test (**j, k**). *p* < 0.05, **p* < 0.01, ***p* < 0.001, ****p* < 0.0001 compared to the WT control group. Source data are provided as a Source Data file.

further analyzed our RNA-seq data from *Soleus* muscle of *Myod* KO and WT littermates fed with high-fat diet (HFD) or standard diet (SD). The data showed the increased mRNA levels of the genes involved in the PC biosynthesis pathway in *Myod* KO mice compared to WT controls in either SD or HFD conditions (Supplementary Fig. 7a–l), further supporting that skeletal muscle of *Myod* KO mice produce more PC than WT littermates. This observation was consistent with the findings of our recent metabolic flux study, in which we demonstrated that the metabolic flux of PC biosynthesis was increased in the oxidative muscle of mice[36].

The increased PC concentration in the skeletal muscle of *Myod* KO mice suggested that PC acts as a sub-class of lipids to induce iWAT browning in *Myod* KO mice. To test this possibility, we isolated stromal vascular fraction (SVF) cells from iWAT of 3-week-old mice and induced the cells to undergo adipocyte differentiation for 6 days. Soybean PC, which contains multiple PC species (18:2/18:2 PC, 16:0/18:2 PC, 18:0/18:2 PC, 16:0/16:0 PC, 16:0/18:1 PC, 18:1/18:0 PC, 18:1/18:1 PC), was applied to differentiated adipocytes at various doses (0.5, 1, and 2 mM) for 12 h. Adipocyte browning was evaluated by measuring Ucp1 expression. Our results confirmed that soybean PC dose-dependently induced Ucp1 expression at the mRNA and protein levels (Fig. 4f, g). PC also dose-dependently induced the additional browning markers, *Elovl3* and *Dio2* (Fig. 4f). Together, our data show that soybean PC functionally induces adipocyte browning in vitro.

Given that the utilized soybean PC was a mixture of various PC species, we asked which PC species can induce adipocyte browning. To this end, we treated the differentiated adipocytes with individual PC species, including 18:2/18:2 PC, 16:0/18:2 PC, 18:0/18:2 PC, 16:0/16:0 PC, 16:0/18:1 PC, 18:1/18:0 PC, 18:1/18:1(6Z) PC, 18:1/18:1(6E) PC, and 18:1/18:1(9E) PC, and then assessed *Ucp1* expression. Intriguingly, 1,2-dilinoleoyl-sn-glycero-3-phosphocholine (18:2/18:2, DLPC), which accounted for 63% of the total soybean PC, functionally induced *Ucp1* expression (Fig. 4h), whereas the other examined PC species did not alter *Ucp1* expression (Fig. 4i, j, Supplementary Fig. 5b–g). Thus, DLPC appears to be a major soybean PC that functions to induce adipocyte browning. When we quantified individual PC species in muscle-derived CM and serum of *Myod* KO mice, we found that the 36:4 PC species, a form of DLPC, was significantly elevated in the muscle-derived CM and sera of *Myod* KO mice compared to those of WT littermates (Supplementary Fig. 5h, i, Supplementary Data. 3, 4). Furthermore, we directly validated that DLPC is a bona fide myolipokine secreted by skeletal muscle cells (Supplementary Fig. 6a, b). These findings together suggest that muscle-secreted DLPC may induce iWAT browning in *Myod* KO mice.

### DLPC prevents HFD-induced obesity

The evidence suggesting that DLPC induces iWAT browning prompted us to assess the ability of DLPC to prevent HFD-induced obesity. To test this, we treated 8-week-old male C57BL/6j mice with various doses of DLPC (50, 100, 200 mg/kg) by intraperitoneal injection (i.p.) for 14 weeks under HFD feeding. Compared to vehicle-control mice, DLPC-treated mice had lower body weights throughout the HFD feeding period (Supplementary Fig. 8a) and showed dose-dependent

decreases in the masses of iWAT and gWAT (Fig. 5a, b). Consistently, the cell sizes of iWAT and gWAT were also dose-dependently decreased in DLPC-treated mice compared to control mice (Fig. 5c).

Metabolic chamber analyses demonstrated that DLPC-treated mice exhibited significantly higher oxygen consumption (Fig. 5d), carbon dioxide production (Fig. 5e), and energy expenditure (Fig. 5f), as well as a decreased respiratory exchange ratio (RER), compared to control mice (Fig. 5g). These changes were accompanied by improvements in glucose tolerance (Fig. 5h, Supplementary Fig. 8b, c) and insulin tolerance (Fig. 5i, Supplementary Fig. 8d, e) in DLPC-treated mice relative to control mice. The improved metabolic profile of DLPC-treated mice was further supported by our observation that the serum levels of total cholesterol (T-CHO) and low-density lipoprotein cholesterol (LDL-C) were decreased (Supplementary Fig. 8f, g). There was no between-group difference in the intake of food (Supplementary Fig. 8h) or water (Supplementary Fig. 8i), indicating that the ability of DLPC to prevent obesity in mice was not due to a decrease in appetite. Mechanistically, we found that DLPC induced the gene expression levels of the browning markers, *Ucp1*, *Prdm16*, and *Cidea*, and the mitochondria-related genes, *Cox7a*, *Cox8b*, *PGC1α*, and *Tfam* (Supplementary Fig. 9a–g). These results collectively demonstrate that DLPC efficiently prevents HFD-induced obesity via induction of iWAT browning in mice.

### DLPC treats obesity in DIO mice

We next examined whether DLPC could alleviate obesity in HFD-induced obese (DIO) mice. 8-week-old male C57BL/6j mice were fed HFD for 8 weeks, and the generated DIO mice were intraperitoneally treated with various doses of DLPC (50, 100, and 200 mg/kg) for an additional 10 weeks of HFD feeding. During the treatment period, the DLPC-treated DIO mice showed a significant loss of body weight, while the control mice continued to gain weight (Supplementary Fig. 10a). Notably, the iWAT and gWAT masses of DLPC-treated DIO mice were remarkably decreased (Fig. 6a, b). The cell size of adipocytes was significantly decreased in the iWAT and gWAT of DLPC-treated DIO mice relative to control mice (Fig. 6c). Consistent with our analysis of the preventive effect of DLPC, we found that DLPC-treated DIO mice showed improved glucose intolerance (Fig. 6d, Supplementary Fig. 10b, c) and insulin sensitivity (Fig. 6e, Supplementary Fig. 10d, e), as well as decreases in the serum levels of T-CHO and LDL-C (Fig. 6f, g). Meanwhile, the food intake (Supplementary Fig. 10f) and water intake (Supplementary Fig. 10g) did not differ between the two groups.

Consistent with our earlier observation, DLPC significantly induced the expression levels of the browning marker genes, *Ucp1*, *Prdm16*, and *Cidea*, and the mitochondria-related genes, *Cox7a*, *Cox8b*, *PGC1α*, and *Tfam*, in the DIO mice (Supplementary Fig. 11a–g). Thus, DLPC appears to be an effective metabolite for treating obesity in DIO mice.

These prevention and treatment results, together with the data showing that muscle-secreted DLPC induces iWAT browning in *Myod* KO mice, lead us to conclude that DLPC is a browning inducer that may be leveraged to ameliorate obesity in mice.

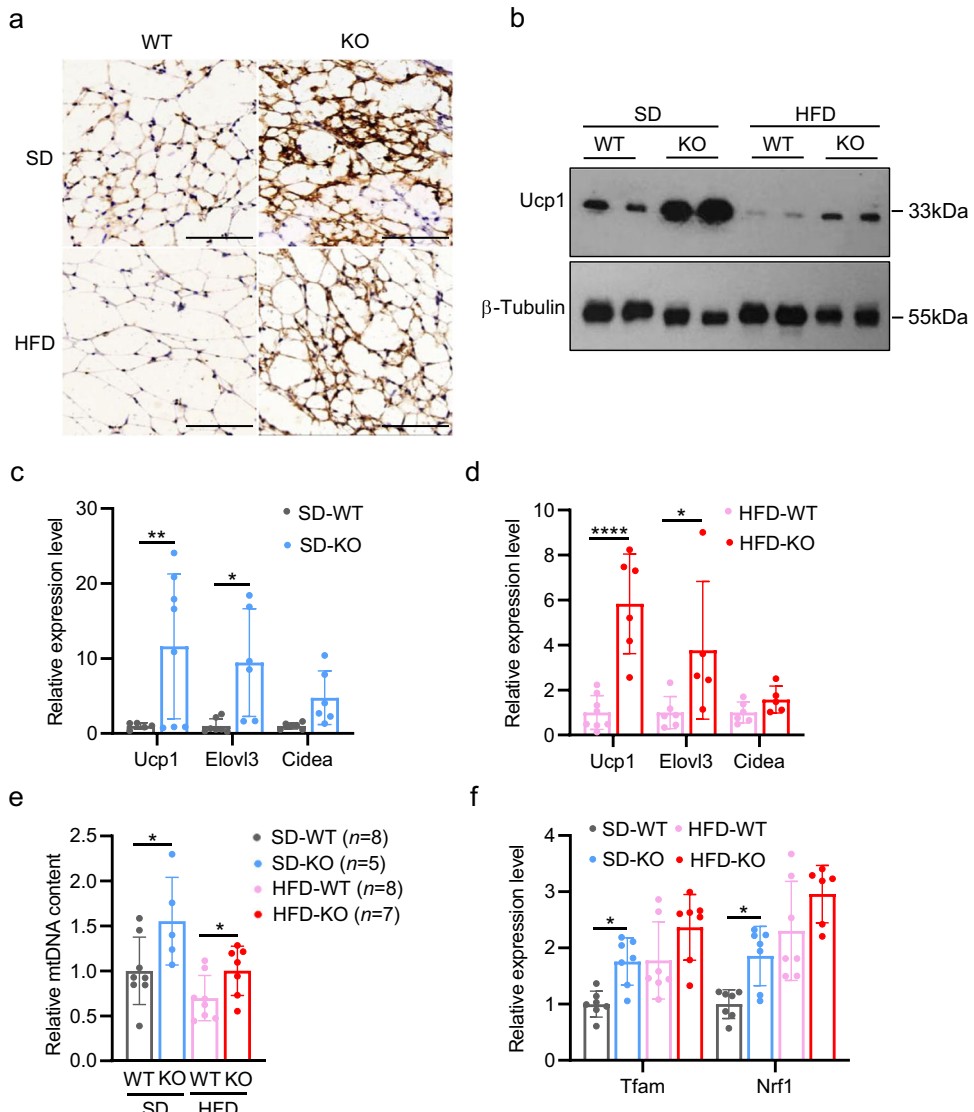

**Fig. 3 | iWAT browning occurs in *Myod* KO mice. a** Representative immunostaining of uncoupling protein 1 (Ucp1) on cryosections of iWAT from *Myod* KO mice and WT littermates fed with SD or HFD, *n* = 3 mice. Scale bar, 100 μm. **b** Western blots showing protein levels of Ucp1 in iWAT from *Myod* KO mice and WT littermates fed with SD or HFD. **c** Relative mRNA levels of browning-related genes in iWAT from *Myod* KO mice and WT littermates fed with SD, as determined by RT-qPCR. For *Ucp1* gene, *n* = 5 (SD-WT), *n* = 8 (SD-KO) mice. *p* = 0.0061 (SD-WT vs. SD-KO); For *Elovl3* gene, *n* = 7 (SD-WT), *n* = 6 (SD-KO) mice. *p* = 0.0293 (SD-WT vs. SD-KO); For *Cidea* gene, *n* = 6 (SD-WT), *n* = 6 (SD-KO) mice. **d** Relative mRNA levels of browning-related genes in iWAT from *Myod* KO mice and WT littermates fed with HFD, as determined by RT-qPCR. For *Ucp1* gene, *n* = 8 (HFD-WT), *n* = 6 (HFD-KO) mice. *p* < 0.0001 (HFD-WT vs. HFD-KO); For *Elovl3* gene, *n* = 6 (HFD-WT), *n* = 5

(HFD-KO) mice. *p* = 0.0173 (HFD-WT vs. HFD-KO); For *Cidea* gene, *n* = 6 (HFD-WT), *n* = 5 (HFD-KO) mice. **e** Quantification of mitochondrial DNA content in iWAT from *Myod* KO mice and WT littermates fed with HFD or SD, *p* = 0.0412 (SD-WT vs. SD-KO), *p* = 0.0430 (HFD-WT vs. HFD-KO). **f** Relative mRNA levels of mitochondrial biogenesis-related genes in iWAT from *Myod* KO mice and WT littermates fed with SD or HFD, as determined by RT-qPCR. For *Tfam* gene, *n* = 7 (SD-WT), *n* = 7 (SD-KO), *n* = 7 (HFD-WT), *n* = 7 (HFD-KO) mice. *p* = 0.0354 (SD-WT vs. SD-KO). For *Nrf1* gene, *n* = 7 (SD-WT), *n* = 7 (SD-KO), *n* = 7 (HFD-WT), *n* = 6 (HFD-KO) mice. *p* = 0.0151 (SD-WT vs. SD-KO). Data are presented as mean ± SD. Significance was assessed by two-way ANOVA (**c-d**, **f**) or two tail Student's *t*-test (**e**). \**p* < 0.05, \*\**p* < 0.01, \*\*\*\**p* < 0.0001 compared to WT control group. Source data are provided as a Source Data file.

## DLPC induces iWAT browning via lipid peroxidation-mediated p38 activation

We next investigated the underlying molecular mechanism by which DLPC induces iWAT browning. As DLPC contains two unsaturated fatty acid chains (18:2/18:2) that are prone to lipid peroxidation[43], we first examined whether DLPC induces lipid peroxidation in adipocytes. To this end, we treated differentiated adipocytes with 0.25 mM DLPC for 30 min in the presence or absence of the lipid peroxidation inhibitor, liproxstatin-1 (Lip-1) and visualized lipid peroxidation with the Bodipy C11 probe (Supplementary Fig. 12a). We found that 0.25 mM DLPC significantly increased the signal intensity of Bodipy C11 (Fig. 7a) and Lip-1 significantly attenuated this DLPC-induced Bodipy C11 signal

(Fig. 7a), suggesting that DLPC induces lipid peroxidation in adipocytes. As lipid peroxidation generally alters the cellular redox state, we performed redox proteomics analysis on adipocytes treated with 0.25 mM DLPC for 30 min in the presence or absence of Lip-1. DLPC significantly increased the number of proteins exhibiting redox state changes and Lip-1 blocked this DLPC-induced protein oxidation (Supplementary Fig. 12b-g, Fig. 7b), further supporting the notion that DLPC induces lipid peroxidation in adipocytes. To further examine whether DLPC-induced *Ucp1* expression depends on lipid peroxidation, we treated differentiated adipocytes with various doses of DLPC in the presence or absence of Lip-1 or Fer-1, which is another inhibitor of lipid peroxidation. Consistent with our earlier observation, DLPC

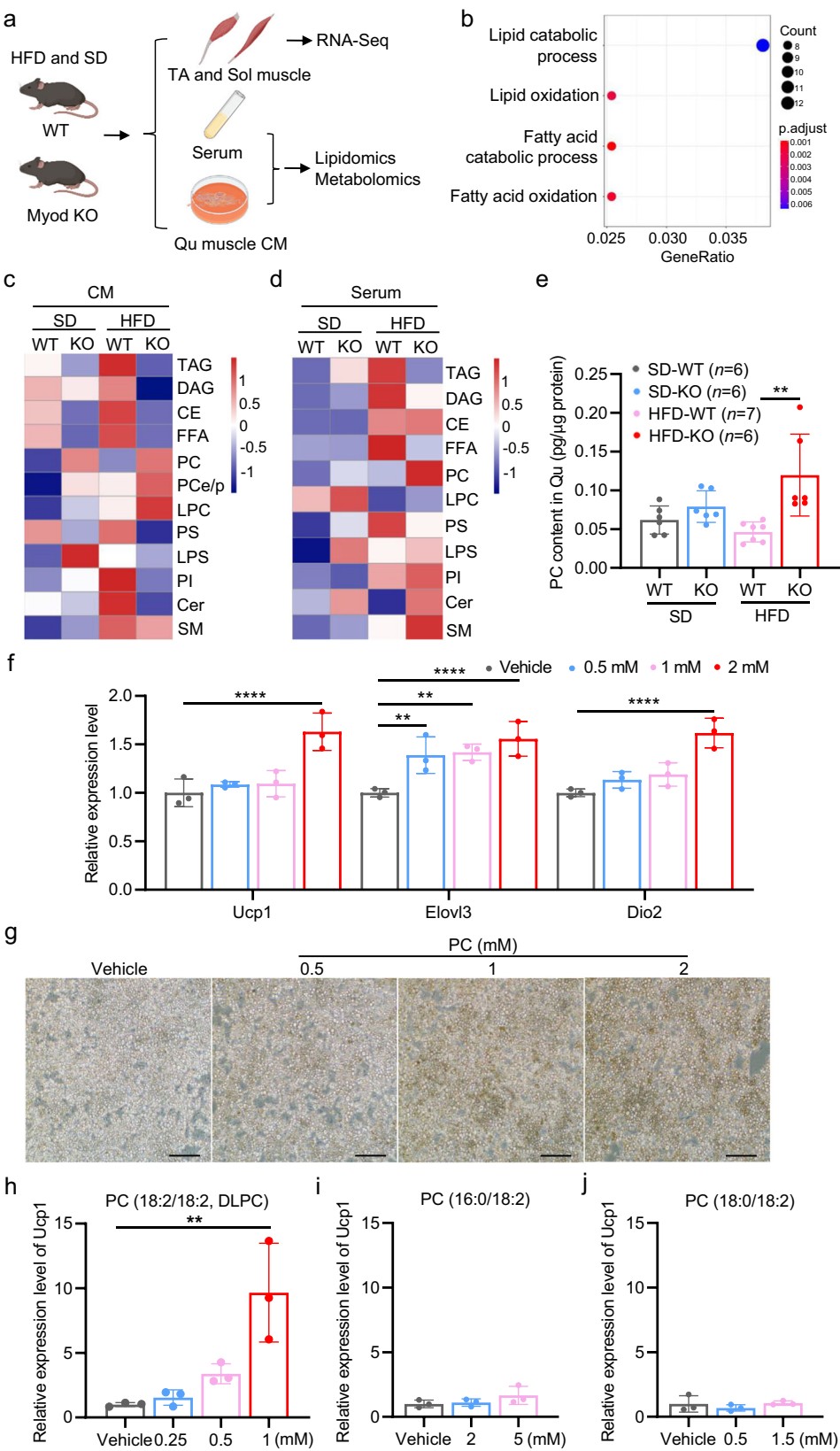

dose-dependently induced *Ucp1* expression (Fig. 7c). Both Lip-1 and Fer-1 completely blocked this DLPC-induced *Ucp1* expression (Fig. 7c). These data indicate that DLPC induces *Ucp1* expression via lipid peroxidation.

To further elucidate the DLPC-induced intracellular signaling responsible for *Ucp1* expression in our system, we performed

phospho-proteomics analysis of adipocytes treated with 0.25 mM DLPC for 30 min in the presence or absence of Lip-1 (Supplementary Fig. 12b–d, Supplementary Fig. 12h-j). The phospho-proteomics data demonstrated that the p38 pathway is involved in DLPC-induced intracellular signaling (Fig. 7d, e). We validated these proteomics data by Western blotting and found that DLPC dose- and time-dependently

**Fig. 4 | DLPC induces iWAT browning in HFD-fed *Myod* KO mice. a** Experimental scheme for using RNA-seq, metabolomics, and/or lipidomics analyses to identify skeletal muscle-secreted lipids or metabolites from *Myod* KO mice fed with HFD for 2 weeks. TA and Sol muscles were collected for RNA-seq. *Quadriceps* (Qu) was used to prepare muscle-conditioned medium (CM) for metabolomics or lipidomics analyses. Sera were collected for metabolomics or lipidomics profiling. Graphs in (**a**) is generated by BioRender (https://www.biorender.com). **b** Enriched GO terms among the skeletal muscle DEGs identified by comparing the *Myod* KO mice and WT littermates described in (**a**). **c** Heatmap showing lipidomics results for muscle-derived CM generated from the mice described in (**a**). **d** Heatmap showing lipidomics results for sera collected from the mice described in (**a**).
**e** Phosphatidylcholine (PC) levels in skeletal muscle (Qu) from the *Myod* KO mice and WT littermates described in (**a**), as measured by a commercial kit, *p* = 0.0044 (HFD-WT vs. HFD-KO) **f** Relative mRNA levels of browning-related genes in iWAT-derived primary adipocytes treated with vehicle or various doses of soybean PC

(0.5, 1, 2 mM) for 12 h, as determined by RT-qPCR. Data are representative of three independent experiments. For *Ucp1* gene, *p* < 0.0001 (Vehicle vs. 2 mM); For *Elovl3* gene, *p* = 0.0032 (Vehicle vs. 0.5 mM), *p* = 0.0016 (Vehicle vs. 1 mM), *p* < 0.0001 (Vehicle vs. 2 mM); For *Dio2* gene, *p* < 0.0001 (Vehicle vs. 2 mM). **g** Representative immunostaining of Ucp1 in iWAT-derived primary adipocytes treated with vehicle or various doses of soybean PC (0.5, 1, 2 mM) for 12 h. Data are representative of three independent experiments. Scale bar, 100 μm. **h–j** Relative mRNA levels of *Ucp1* in iWAT-derived primary adipocytes treated with vehicle or various doses of DLPC (0.25, 0.5, 1 mM) (**h**), 16:0/ 18:2 PC (2, 5 mM) (**i**), 18:0/ 18:2 PC (0.5, 1.5 mM) (**j**) for 12 h, as determined by RT-qPCR. *p* = 0.0017 (Vehicle vs. 1 mM DLPC). Data are representative of three independent experiments. Data are presented as mean ± SD. Significance was assessed by one-way ANOVA (**h–j**), two-way ANOVA (**f**), or two-tail Student's *t*-test (**e**). \*\**p* < 0.01, \*\*\*\**p* < 0.0001 compared to the vehicle control group. Source data are provided as a Source Data file.

---

activated p38 phosphorylation in adipocytes (Fig. 7f, g). Lip-1 completely blocked DLPC-induced p38 phosphorylation (Fig. 7h), indicating that DLPC activates p38 signaling via lipid peroxidation. To further examine whether DLPC-induced *Ucp1* expression depends on lipid peroxidation-mediated p38 phosphorylation, we treated differentiated adipocytes with DLPC in the presence or absence of Lip-1 or the p38 inhibitor, BIRB796. We found that both Lip-1 and BIRB796 completely blocked DLPC-induced *Ucp1* gene expression (Fig. 7i). Together, these findings led us to conclude that DLPC induces adipocyte browning via lipid peroxidation-mediated p38 activation in vitro.

Finally, we examined DLPC-induced p38 signaling in vivo. We i.p. injected C57BL/6j mice with various doses of DLPC (50, 100, and 150 mg/kg) and, after 30 min, collected iWAT tissues for Western blotting analysis. We found that DLPC dose-dependently activated p38 phosphorylation in the iWAT of mice (Fig. 7j), and this effect was blocked by Lip-1 (Fig. 7k). Together, our results provide mechanistic support for the notion that DLPC induces iWAT browning via lipid peroxidation-mediated p38 activation in vitro and in mice.

In sum, we herein used *Myod* KO and HFD-induced obesity mouse models and multi-omics approaches to identify DLPC as a browning inducer that mediates whole-body metabolic homeostasis via lipid peroxidation-mediated p38 activation in mice.

## Discussion

MyoD is a muscle-specific TF that is well known to function in myogenic lineage specification during embryonic skeletal muscle development[21–23]. It is also expressed in mature skeletal muscle, where it functions in controlling fiber type switch[30,31] and regulating the metabolic capacity of mature skeletal muscle to provide sufficient energy for muscle contraction[34]. Using *Myod* KO mice as a genetic model, we herein explored the functional significance of skeletal muscle in regulating the whole-body metabolic balance. Firstly, the elevated oxidative metabolism in skeletal muscle of *Myod* KO mice generally contributed to systemic homeostasis. Notably, we uncovered the role of MyoD in controlling whole-body energy homeostasis by communicating with fat via muscle-released DLPC in mice. This is the genetic study to identify a metabolic function of MyoD in regulating the crosstalk between muscle and fat in mice. Notably, our findings provide molecular insights that advance our understanding of the physiological and pathological roles of skeletal muscle in regulating whole-body energy homeostasis in mice and (by extension) humans.

An important question remaining to be answered is: What is the molecular mechanism underlying the function of muscle in orchestrating systemic metabolism in vivo? Skeletal muscle is an important endocrine organ, and a class of muscle-released circulating cytokines called "myokines" have been shown to act as signaling molecules and mediate systemic metabolism in vivo[19,44]. Myokines reportedly play significant biological roles in regulating whole-body energy

homeostasis by actively communicating with other metabolic organs, such as fat and liver[19,20]. Given that the skeletal muscle is also a metabolic organ and its metabolic capacity is closely related to the metabolic state of the body, it is conceivable that skeletal muscle might orchestrate whole-body energy homeostasis by secreting signaling metabolites.

In this work, we searched for skeletal muscle-secreted metabolites that could control whole-body energy homeostasis through metabolic communication between muscle and fat in mice. By using *Myod* KO mice and multi-omics approaches, we identified muscle-secreted DLPC as a signaling metabolite that communicates with fat to orchestrate whole-body metabolic homeostasis in mice. Functionally, we demonstrated that DLPC effectively ameliorates HFD-induced obesity in mice. Mechanistically, we found that DLPC protects mice against obesity by inducing iWAT browning through lipid peroxidation-mediated p38 activation (Fig. 8). The key finding is our discovery of a novel ability of the muscle-secreted metabolite, DLPC, to induce browning and treat obesity in mice.

We herein reported the first muscle-secreted lipokine, DLPC, and show that it acts as a signaling metabolite to induce iWAT browning both in vitro and in vivo. Lipokines are circulating lipid species that were recently shown to act as signaling molecules in mediating systemic metabolism[11,12,14,45–50]. Most of the lipokines reported to date, including palmitoleate, 12,13-diHOME, and fatty acid−hydroxy−fatty acids (FAHFA) species, are secreted from adipose tissues[14,48,49]. However, no previous report assessed whether muscle could secrete a lipokine. To our knowledge, this is the first report of a muscle-secreted circulating lipokine. Our findings could lay the foundation for the identification of other types of muscle-secreted lipokines and significantly promote studies on the ability of muscle-secreted lipokines to act as signaling metabolites in mediating communication between metabolic tissues/organs.

Phosphatidylcholine (PC) is a phospholipid that is well known to be a major component of cell membranes[51]. PC generally accounts for 40%-50% of the total membrane phospholipids, but its abundance varies across different types of membranes[52]. In addition to this classic function, PC also functions in regulating cellular metabolism. Chakravarthy et al.[53] reported that 16:0/18:1 PC functions as a physiologically relevant endogenous PPARα ligand to induce PPARα-dependent gene expression and decrease hepatic steatosis in mice. The 12:0/12:0 PC species might act as an agonist of the orphan nuclear receptor LRH-1 to induce bile acid biosynthetic enzymes and decrease hepatic steatosis in mouse liver[54]. Diurnal hepatic PPARδ-regulated 18:0/18:1 PC, as a serum lipid, mediates liver-muscle crosstalk to reduce postprandial lipid levels and increase fatty acid use through muscle PPARα in mice[55]. In human, the serum concentration of PC is negatively associated with cellular metabolism, human obesity and T2D[56]. Palumbo et al.[57] injected PC into human adipose tissue and found that PC could significantly reduce adipose tissue mass. Other studies

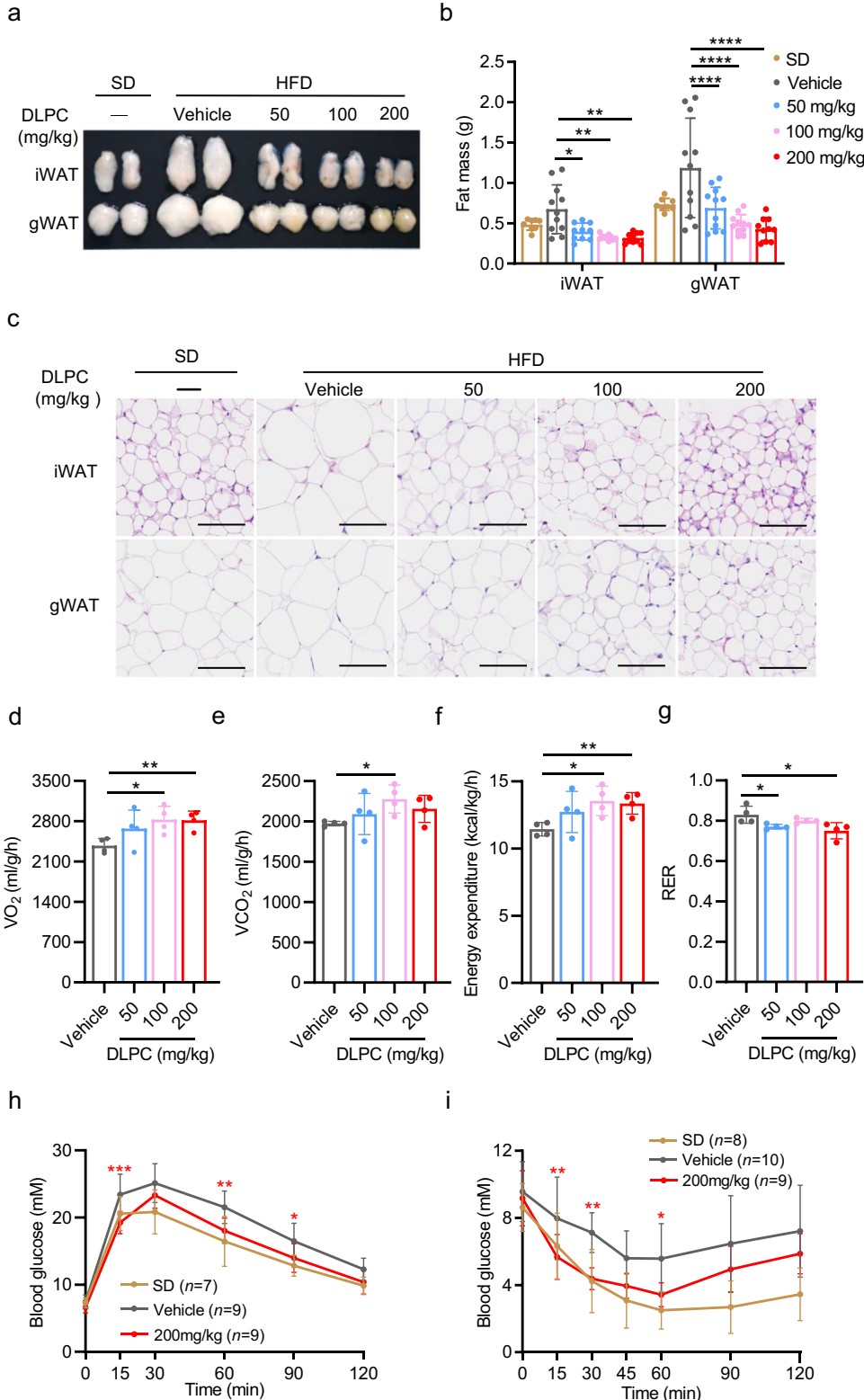

showed that PC significantly prevents body weight gain, decreases fat mass, and mediates adipose tissue lipolysis in mice[58,59]. The PC used in the early studies in adipose tissues comprised mixtures of various PC species; thus it is unclear which PC species are particularly effective and what molecular mechanisms form the basis for the ability of a given PC to regulate cellular metabolism in adipose tissues. In this study, though we could not rule out the possibility that additional types of PC might regulate whole body energy homeostasis, we

provide the evidence indicating that the single PC species, 18:2/18:2 DLPC, has a signal transduction function in orchestrating whole-body metabolic homeostasis via mediating muscle-fat crosstalk. Our further mechanistic investigations revealed that DLPC ameliorates HFD-induced obesity by inducing iWAT browning via lipid peroxidation-mediated p38 activation in mice.

Together, our findings uncover systems-level molecular mechanisms through which the first reported muscle-secreted

**Fig. 5 | DLPC prevents HFD-induced obesity. a** Fat pads were collected from C57BL/6j mice that were fed with SD or HFD and simultaneously intraperitoneally (i.p.) administered with vehicle or various doses of DLPC (50, 100, 200 mg/kg) daily for 14 weeks. **b** Masses of iWAT and gWAT from the mice described in (**a**); For iWAT mass, $n = 8$ (SD), $n = 11$ (Vehicle), $n = 10$ (50 mg/kg), $n = 10$ (100 mg/kg), $n = 10$ (200 mg/kg) mice. $p = 0.0491$ (Vehicle vs. 50 mg/kg), $p = 0.0091$ (Vehicle vs. 100 mg/kg), $p = 0.0069$ (Vehicle vs. 200 mg/kg). For gWAT mass, $n = 8$ (SD), $n = 11$ (Vehicle), $n = 11$ (50 mg/kg), $n = 11$ (100 mg/kg), $n = 10$ (200 mg/kg) mice. $p < 0.0001$ (Vehicle vs. 50 mg/kg), $p < 0.0001$ (Vehicle vs. 100 mg/kg), $p < 0.0001$ (Vehicle vs. 200 mg/kg). **c** Representative H&E-stained sections of iWAT and gWAT from the mice described in (**a**). Scale bar, 100 μm. **d–g** $O_2$ consumption (d) and $CO_2$ production (e), energy expenditure (f) and respiratory exchange ratio (RER) (g) by the mice described in (**a**), as measured by metabolic chamber analysis, $n = 4$ mice. In (**d**), $p = 0.0151$ (Vehicle vs. 100 mg/kg), $p = 0.0046$ (Vehicle vs. 200 mg/kg). In (**e**), $p = 0.0136$ (Vehicle vs. 100 mg/kg). In (**f**), $p = 0.0131$ (Vehicle vs. 100 mg/kg), $p = 0.0065$ (Vehicle vs. 200 mg/kg). In (**g**), $p = 0.0364$ (Vehicle vs. 50 mg/kg), $p = 0.0343$ (Vehicle vs. 200 mg/kg). **h** GTT performance of the mice described in (**a**), $p = 0.0002$ (15 min), $p = 0.0021$ (60 min), $p = 0.0310$ (90 min). **i** ITT performance of the mice described in (**a**), $p = 0.0083$ (15 min), $p = 0.0014$ (30 min), $p = 0.0140$ (60 min). Data are presented as mean ± SD. Significance was assessed by two-way ANOVA (**b, h–i**) or two tail Student's $t$-test (**d–g**). *$p < 0.05$, **$p < 0.01$, ***$p < 0.001$, ****$p < 0.0001$ compared to vehicle control group. Source data are provided as a Source Data file.

lipokine, DLPC, contributes to maintaining metabolic homeostasis. Going forward, comprehensively understanding the regulation of DLPC synthesis and secretion in skeletal muscle could critically contribute to efforts aimed at developing preventative and therapeutic strategies to treat obesity and its myriad metabolic consequences in humans.

## Limitation of the study
Although our findings reveal the physiological and pathological significance of the muscle-released DLPC acting as a signaling metabolite to control whole-body energy homeostasis by inducing iWAT browning in mice, there are still several limitations in term of mechanistic insights into DLPC functions. How does MyoD regulate DLPC biosynthesis and secretion in skeletal muscle? What's direct binding protein(s) or receptor(s) of DLPC in adipocytes? To what extent the muscle and adipose browning contribute to the overall metabolic balance? We are dedicated to accumulating more robust evidence to shed light on those questions in our future research endeavors.

## Methods
### Mouse models
All animal procedures were approved by the Animal Ethics Committee of Institute of Basic Medical Sciences, Chinese Academy of Medical Sciences (Beijing). Animals were euthanized by cervical dislocation and ARRIVE guidelines were followed. Mice were housed in a pathogen-free facility and had free access to water and standard rodent chow under the following conditions: 21 °C ambient temperature, 50–60% humidity, 12 h dark/light cycle. *Myod*-knockout (KO) mice (#002523) were obtained from the Jackson Laboratory. C57BL/6j male mice were purchased from Beijing HFK Bioscience Co., Ltd. The floxed-*Myod* mice in the C57BL/6j background were generated by the Model Animal Research Center of Nanjing University. For studies with specific diets, 8-week-old male mice were fed with high-fat diet (HFD, 45% fat, Medicine Diets, MD12032) or standard diet (SD, 10% fat, Medicine Diets, MD12031).

### Prevention and treatment of HFD-induced obesity with DLPC
For the prevention study, 8-week-old male C57BL/6j mice were randomly divided into 5 groups ($n = 6$–12/group) and fed with HFD with SD as control. During HFD feeding, DLPC (50, 100, 200 mg/kg) or vehicle was intraperitoneally (i.p.) injected to mice once daily for 14 weeks. For the treatment experiments, 8-week-old male C57BL/6j mice were fed with HFD for 8 weeks to induce obesity or fed with SD as control. The obese mice were randomly divided into five groups ($n = 6$–12/group) and i.p. injected with DLPC (50, 100, 200 mg/kg) or vehicle once daily for 10 weeks. After the final injection, mice underwent glucose and insulin tolerance tests and were then sacrificed for sampling.

### Glucose and insulin tolerance tests
For the glucose-tolerance test (GTT), overnight-fasted mice were i.p. injected with glucose (2 mg/g body weight). For the insulin-tolerance

tests (ITT), mice were fasted for 5 h and then i.p. injected with 0.75 U insulin/g body weight (Novolin). Blood glucose was determined with a Lifescan One Touch glucometer.

### Metabolic-chamber analysis
Metabolic phenotyping of *Myod* KO and wild-type littermates or DLPC-treated mice was performed using Columbus × Oxymax / CLAMS metabolic-chamber analysis at the animal center of Institute of Basic Medical Sciences, Chinese Academy of Medical Sciences.

### Serum lipid measurement
Serum levels of various lipids, including cholesterol (Chol), high-density lipoprotein cholesterol (HDL-C), low-density lipoprotein cholesterol (LDL-C), and triglycerides (TG) were determined by using a three-agent kit from Nanjing Jiancheng Biomedical Company (Nanjing, Jiangsu, China).

### Primary adipocyte isolation and cell culture
Stromal vascular fraction (SVF) cells were isolated from the iWAT of 3- to 4-week-old C57BL/6j mice and cultured in Dulbecco's Modified Eagle Medium/Nutrient Mixture F-12 (DMEM/F-12, GIBCO) supplemented with 10% calf serum (NCS, Capricorn Scientific) in 5% $CO_2$ at 37 °C. Cells with 90% confluence were passaged and seeded to 12-well plates. When the passaged cells reached 60 - 70% confluence, they were fed DMEM/F12 medium supplemented with 10% fetal bovine serum (FBS, Capricorn Scientific) and 1% penicillin-streptomycin for 24 h. To induce differentiation, the cells were switched to DMEM containing 10% FBS, 0.5 mM 3-isobutyl-1-methylxanthine, 1 mM dexamethasone, and 10 mg/mL insulin. After 2 days, the cells were switched to DMEM containing 10% FBS and 10 mg/mL insulin, and culture was continued for an additional 6-8 days until lipid droplets appeared.

To investigate the effects of DLPC, differentiated adipocytes were treated with vehicle or DLPC (0.25, 0.5, 1 mM) for 0.5–12 h and then adipocyte browning was evaluated by measuring Ucp1 expression. For inhibitor treatment, differentiated adipocytes were pre-incubated with 50 μM liproxstatin-1 (Selleck, 950455-15-9) or 10 μM BIRB796 (Selleck, S1574) for 30 min and then treated with DLPC for 12 h.

### Primary myoblast isolation, culture, and differentiation
Primary myoblasts were isolated from hind limb skeletal muscles of the floxed *Myod* (*Myod*^f/f) mice and WT littermates at 2 - 3-week-old[60]. Briefly, the muscle were minced and digested in a mixture of type II collagenase and dispase. Cells were filtered from debris, centrifuged, and purified to eliminate fibroblasts by differential attachment for 30 min. The obtained cells were cultured in growth medium (F-10 Ham's medium supplemented with 20% fetal bovine serum, 10 ng/ml basic fibroblast growth factor, 1% antibiotics) on collagen-coated cell culture plates at 37 °C in 5% $CO_2$. For induced differentiation of primary myoblasts, the cells with 70–80% confluence were transferred to Dulbecco's modified Eagle's medium (Gibco, Cat.N: C11995500BT) containing 2% horse serum (Hyclone, Cat.N: SH30074.03) and 1% penicillin and streptomycin, and then cultured for 48 h. For

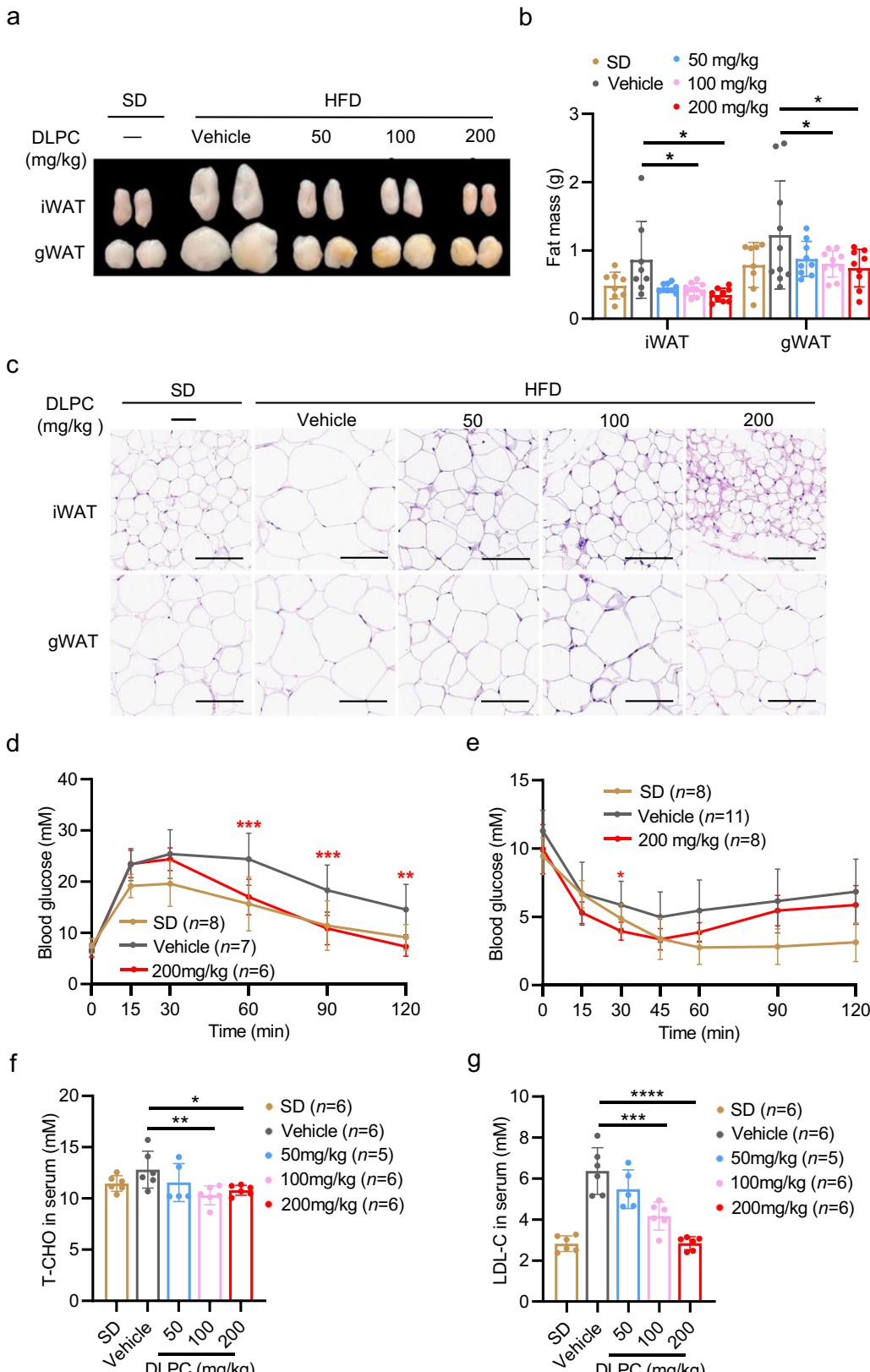

mitochondrial respiration assay, the 48 h-differentiated myotubes from *Myod*fl/fl mice and WT controls were infected with adenovirus-Cre (ad-Cre) or adenovirus-EGFP as control (ad-Ctrl), and further cultured for additional 48 h. Then, the myotubes were subjected to mitochondrial respiration assay, quantitative PCR (qPCR) analyses and immunofluorescent staining.

## Seahorse XFe24 measurements

Mitochondrial respiration of myotubes was measured as oxygen consumption rate (OCR), using an oxygen-controlled XFe24 Analyzer (Agilent Technologies). Primary myoblasts were cultured in Seahorse XF cell culture microplates (Agilent Technologies) and induced differentiation for 96 h. Before the mitochondrial respiration test, the

**Fig. 6 | DLPC treats obesity in DIO mice. a** Fat pads from HFD-induced obese (DIO) C57BL/6j mice that were i.p. administered with vehicle or various doses of DLPC (50, 100, 200 mg/kg) daily for 10 weeks $n = 12$ mice. Mice fed with SD served as controls. **b** Masses of iWAT and gWAT from the mice described in (**a**); For iWAT mass, $n = 8$ (SD), $n = 8$ (Vehicle), $n = 9$ (50 mg/kg), $n = 10$ (100 mg/kg), $n = 9$ (200 mg/kg-iWAT) mice. $p = 0.0444$ (Vehicle vs. 100 mg/kg), $p = 0.0162$ (Vehicle vs. 200 mg/kg). For gWAT mass, $n = 8$ (SD), n = 10 (Vehicle), $n = 9$ (50 mg/kg), $n = 10$ (100 mg/kg), $n = 9$ (200 mg/kg) mice. $p = 0.0383$ (Vehicle vs. 100 mg/kg), $p = 0.0172$ (Vehicle vs. 200 mg/kg). **c** Representative H&E-stained sections of iWAT and gWAT

from the mice described in (**a**). Scale bar, 100 μm. **d** GTT performance of the mice described in (**a**); $p = 0.0008$ (60 min), $p = 0.0008$ (90 min), $p = 0.0011$ (120 min). **e** ITT performance of the mice described in (**a**); $p = 0.0234$ (30 min). **f** T-CHO in sera from the mice described in (**a**), $p = 0.0079$ (Vehicle vs. 100 mg/kg), $p = 0.0388$ (Vehicle vs. 200 mg/kg). **g** LDL-C in sera from the mice described in (**a**), $p = 0.0001$ (Vehicle vs. 100 mg/kg), $p < 0.0001$ (Vehicle vs. 200 mg/kg). Data are presented as mean ± SD. Significance was assessed by one-way ANOVA (**f**, **g**) or two-way ANOVA (**b**, **d**, **e**). *$p < 0.05$, **$p < 0.01$, ***$p < 0.001$, ****$p < 0.0001$ compared to vehicle control group. Source data are provided as a Source Data file.

sensor cartridge was hydrated in Seahorse XF Calibrant (Agilent Technologies) overnight at 37 °C in a $CO_2$ free incubator. The next day, DMEM XF assay media (with 1 mM pyruvate, 2 mM glutamine and 10 mM glucose, pH = 7.4, Agilent Technologies) was prepared, and the myotubes were incubated in DMEM XF assay media at 37 °C in a $CO_2$ free incubator for 1 h. The mitochondrial respiration test utilizes sequential injections of oligomycin (1.5 μM, Alicelligent, ALS22012), Carbonyl cyanide − 4 - (trifluoromethoxy) - phenylhydrazone (2 μM FCCP, Alicelligent, ALS22012), and rotenone/antimycin A in combination (1 μM, Alicelligent, ALS22012). The data were analyzed using WAVE software (Version 2.6.1.53, Agilent Technologies).

### Bodipy C11 staining and flow cytometry
Adipocytes were treated with vehicle or 0.25 mM DLPC for 3 h in the presence or absence of the lipid peroxidation inhibitors, Lip-1. After treatment, adipocytes were incubated in fresh culture medium containing 1 μM of the Bodipy C11 (Invitrogen) at 37 °C for 30 min. Then the cells were washed with PBS twice and digested with Trypsin. The digested cells were resuspended in PBS for flow cytometry. Data were analyzed using FlowJo (Version 10.8.1) and BD Accuri C6 Plus Software (Version 1.0.23.1).

### Lipid and metabolite extraction
Lipids were extracted from serum samples using a modified method of Bligh and Dyer[61]. Briefly, 750 μL of chloroform:methanol:MilliQ $H_2O$ (3:6:1; v/v/v) containing 0.1% (w/v) butylated hydroxytoluene was added to 20 μL of serum and the mixture was incubated at 1500 rpm for 1 h at 4 °C. Phase separation was induced by the addition of 350 μL deionized water and 250 μL chloroform. The samples were centrifuged and the lower (lipid-containing) organic phase was extracted into a clean tube. Further lipid extraction was performed by adding 450 μL chloroform to the remaining aqueous phase and collecting the lower organic phase following centrifugation. The lipid extracts were pooled into a single tube and dried in a SpeedVac under OH mode. For lipid extraction of culture medium, 4 ml chloroform:methanol (2:1; v/v) was mixed with 1 ml culture of medium and incubated at 1500 rpm for 1 h at 4°C. The sample was centrifuged and the lower organic phase was extracted into a NOV tube using a glass micropipette, dried in a SpeedVac under OH mode, and stored at −80 °C until further analysis. The upper (polar metabolite-containing) aqueous-methanol phase was dried in the SpeedVac under $H_2O$ mode and used for metabolomics analyses.

### Lipidomics analyses
Lipidomic analyses were conducted at LipidALL Technologies using an ExionLC-AD coupled with a Sciex QTRAP 6500 PLUS[62]. The detailed methodologies, including MRM transitions, liquid chromatography gradients, and MS source parameters[63] were: CUR: 25, TEM 400 °C, GS1 35, GS2 35. Individual classes of polar lipids were separated by normal phase (NP)-HPLC using a TUP-HB silica column (i.d. 150 × 2.1 mm, 3 μm) and the following conditions: mobile phase A (chloroform:methanol:ammonium hydroxide, 89.5:10:0.5) and mobile phase B (chloroform:methanol:ammonium hydroxide:water, 55:39:0.5:5.5). The gradient started with 2% B that was maintained for 2 min, which

was then increased steadily to 40% B over the next 1 min, and maintained at 40% B for 6.5 min. % B was further increased to 100% from 9.5 min to 11 min, and maintained at 100% B for 5 min, before returning to 2% B over 1 min. The column was equilibrated at 2% B for 3 min prior to the next injection. The MRM transitions were set for comparative analysis of various polar lipids. Individual lipid species were quantified by referencing to spiked internal standards. DMPC, DMPE, d31-PS(d31-16:0/18:1), PA(17:0/17:0), DMPG, diC8-PI, Cer d18:1/17:0, SM d18:1/12:0, C8-GluCer, C8-GalCer, $d_3$-LacCer d18:1/16:0, Gb3 d18:1/17:0, C17-LPC, C17-LPE, C17-LPI, C17-LPA, C17-LPS, C17-LPG, d17:1 Sph, d17:1 S1P, and $d_3$-C16:0-carnitine were obtained from Avanti Polar Lipids. GM3-d18:1/18:0-d3 was purchased from Matreya LLC. Free fatty acids were quantitated using $d_{31}$-C16:0 (Sigma-Aldrich) and $d_8$-C20:4 (Cayman Chemicals). Glycerol lipids, including diacylglycerols (DAG) and triacylglycerols (TAG), were quantified using a modified version of reverse-phase HPLC/MRM. Separation of neutral lipids was achieved on a Phenomenex Kinetex-C18 column (i.d. 4.6 × 100 mm, 2.6 μm) using an isocratic mobile phase containing chloroform:methanol:0.1 M ammonium acetate at 100:100:4 (v/v/v) at a flow rate of 300 μL for 10 min. Levels of short-, medium-, and long-chain TAG were calculated by referring to spiked internal standards of TAG(14:0)3-$d_5$, TAG(16:0)3-$d_5$, and TAG(18:0)3-$d_5$ (all from CDN isotopes). DAG were quantified using $d_5$-DAG17:0/17:0 and $d_5$-DAG18:1/18:1 as internal standards (Avanti Polar Lipids). Free cholesterols and cholesteryl esters were analyzed under atmospheric pressure chemical ionization (APCI) mode on a Jasper HPLC coupled to a Sciex 4500 MD, as described previously, using $d_6$-cholesterol and $d_6$-C18:0 cholesteryl ester (CE) (CDN isotopes) as internal standards. Comprehensive methodological details including the library of MRM transitions had been reported in a preceding publication[63].

### Metabolomics analyses
Polar metabolites were derivatized with phenylhydrazine to increase the stability of alpha-keto acids[64]. Metabolite extracts were reconstituted in 2% acetonitrile (ACN) in water and subjected to LC-MS analysis on ThermoFisher Dionex UltiMate 3000 HPLC system coupled to Q-Exactive mass spectrometer[65]. Polar metabolites were separated on a Waters ACQUITY HSS-T3 column (3.0 × 100 mm, 1.8 μm). The temperatures of the column and auto-sampler were maintained at 40 °C and 4 °C, respectively. The injection volume was 5 μL, and flow rate was at 0.35 mL/min. Mobile phases A and B were 0.1% formic acid in de-ionized water and ACN, respectively. The linear gradient was 1% B at 1 min, 40% B at 4 min, 50% B at 5 min, 65% B at 8 min, 76% B at 10 min, 100% B at 16 min, 100% B at 21 min, 1% B at 21.1 min, and 1% B at 25 min. MS analysis was conducted with an HESI probe, controlled by the Xcalibur 2.3 software (Thermo Fisher Scientific, Waltham, MA). The MS parameters for detection were as follows: sheath gas flow rate, 40 PSI; auxiliary gas flow rate, 11 Arb; sweep gas flow rate, 0; full MS scan mode was used and scan ranges were 60-600 m/z; spray voltage, 3.5 kV for positive mode and 3.2 kV for negative mode; capillary temperature, 350 °C; s-lens RF level, 55; auxiliary gas heater temperature, 220 °C. High-purity nitrogen (N2) was used as the nebulizing gas and the collision gas for higher energy collisional dissociation. The Q-Orbitrap performs data-dependent scans in full MS/dd-MS2 to obtain product

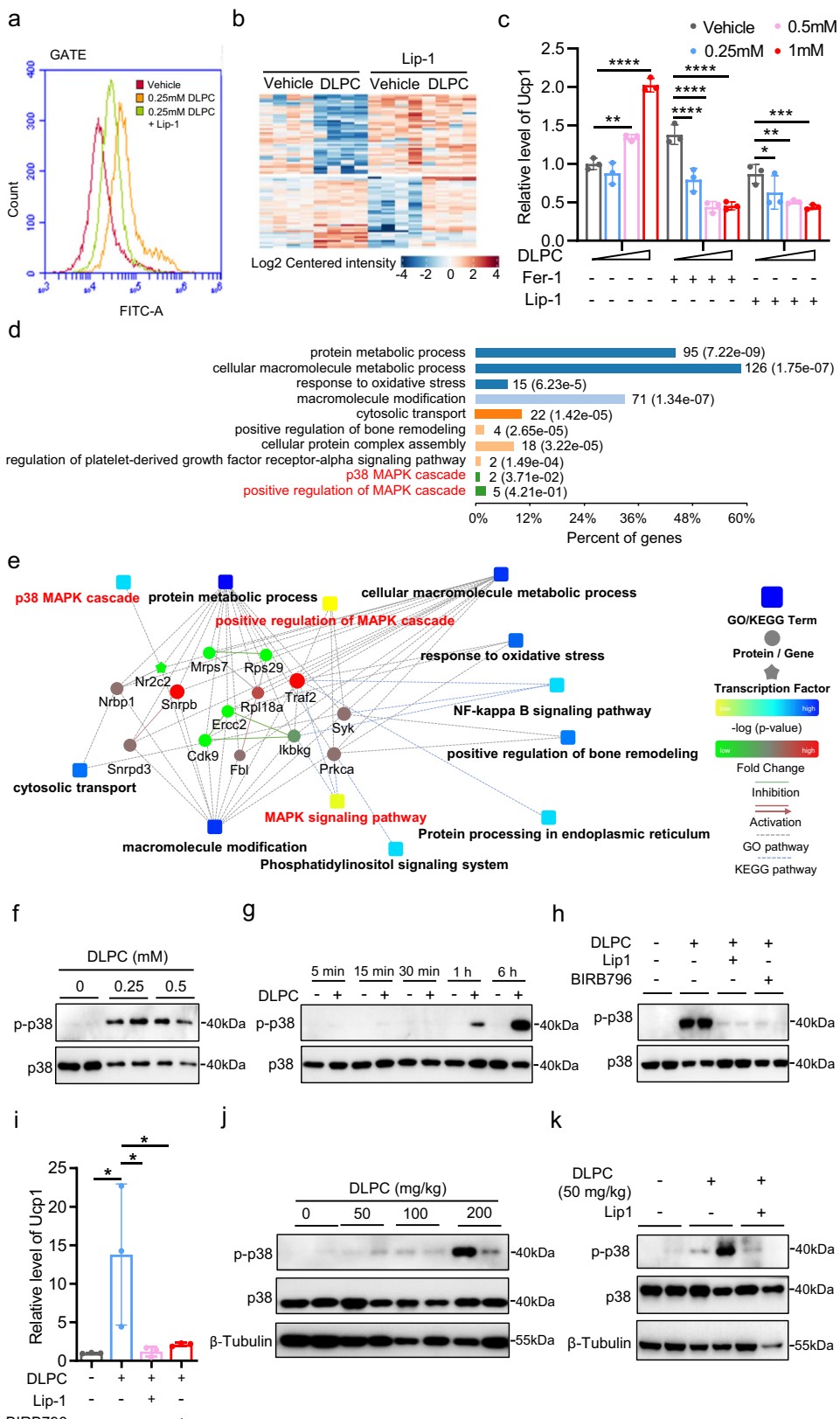

ion information with normalized collision energy (NCE), and the mass resolution was set at 17,500 FWHM (m/z 200) and NCE 35%. High-resolution MS spectra, isotope abundance ratios, MS/MS, the Human Metabolome database (HMDB) at http://www.hmdb.ca/, and comparisons with reference standard compounds were utilized for metabolite identification.

A cocktail of isotopically-labeled internal standards (IS) purchased from Cambridge Isotope Laboratories were spiked into the samples for metabolite quantitation, including L-Tryptophan-d$_8$, L-Isoleucine-d$_{10}$, L-Methionine-d$_3$, L-Valine-d$_8$, L-Proline-d$_7$, L-Alanine-d$_4$, DL-Serine-d$_3$, L-Glutamine-d$_5$, Glycine-d$_2$, L-Aspartic acid-d$_3$, L-Arginine-d$_7$, L-Glutamate-d$_5$, L-Lysine-d$_9$, L-Histidine-$^{13}$C$_6$, Taurine-$^{13}$C$_2$, L-lactate-$^{13}$C$_3$,

**Fig. 7 | DLPC induces iWAT browning via lipid peroxidation-mediated p38 activation. a** FACS profiles of Bodipy C11-stained iWAT-derived adipocytes treated with vehicle or 0.25 mM DLPC for 30 min in the presence or absence of the lipid peroxidation inhibitor, liproxstatin-1 (Lip-1). **b** Heatmap showing redox proteomics results from iWAT-derived adipocytes treated with vehicle or 1 mM DLPC for 30 min in the presence or absence of Lip-1. **c** Relative mRNA levels of *Ucp1* in iWAT-derived primary adipocytes treated with vehicle or various doses of DLPC (0.25, 0.5, 1 mM) for 12 h in the presence or absence of the lipid peroxidation inhibitors, Lip-1 or Fer-1. For DLPC group, $p = 0.0025$ (Vehicle vs. 0.5 mM), $p < 0.0001$ (Vehicle vs. 1 mM); For DLPC + Fer-1 group, $p < 0.0001$ (Vehicle vs. 0.25 mM), $p < 0.0001$ (Vehicle vs. 0.5 mM), $p < 0.0001$ (Vehicle vs. 1 mM); For DLPC + Lip-1 group, $p = 0.0345$ (Vehicle vs. 0.25 mM), $p = 0.0014$ (Vehicle vs. 0.5 mM), $p = 0.0002$ (Vehicle vs. 1 mM). Data are representative of three independent experiments. **d** Pathway enrichment from phospho-proteomics analysis of iWAT-derived adipocytes treated with vehicle or 1 mM of DLPC for 30 min in the presence or absence of Lip-1. **e** Network showing the proposed involvement of p38 and the MAPK

cascade in the DLPC-mediated signaling of the iWAT-derived adipocytes described in (**d**). **f–h** Representative Western blots showing p38 phosphorylation in iWAT-derived adipocytes treated with vehicle or various doses of DLPC (0.25, 0.5 mM) for 12 h (**f**), or 1 mM DLPC for the indicated durations (5 min, 15 min, 30 min, 1 h, 6 h) (**g**), or 1 mM DLPC for 12 h in the presence or absence of Lip-1 or BIRB796 (an inhibitor of p38) (**h**). **i** Relative mRNA levels of *Ucp1* in iWAT-derived adipocytes treated as described in (**h**), as determined by RT-qPCR. $p = 0.0228$ (Vehicle vs. DLPC), $p = 0.0245$ (DLPC vs. DLPC+Lip-1), $p = 0.0352$ (DLPC vs. DLPC + BIRB796). Data are representative of three independent experiments. **j, k** Representative Western blots showing p38 phosphorylation in iWAT from C57BL/6j mice i.p. administered with vehicle or various doses of DLPC (50, 100, 200 mg/kg) for 30 min (**j**) or 50 mg/kg DLPC for 30 min in the presence or absence of Lip-1 (**k**). β-Tubulin served as an equal loading control. Data are presented as mean ± SD. Significance was assessed by one-way ANOVA (**i**) or two-way ANOVA (**c**). *$p < 0.05$, **$p < 0.01$, ***$p < 0.001$, ****$p < 0.0001$ compared to vehicle control group. Source data are provided as a Source Data file.

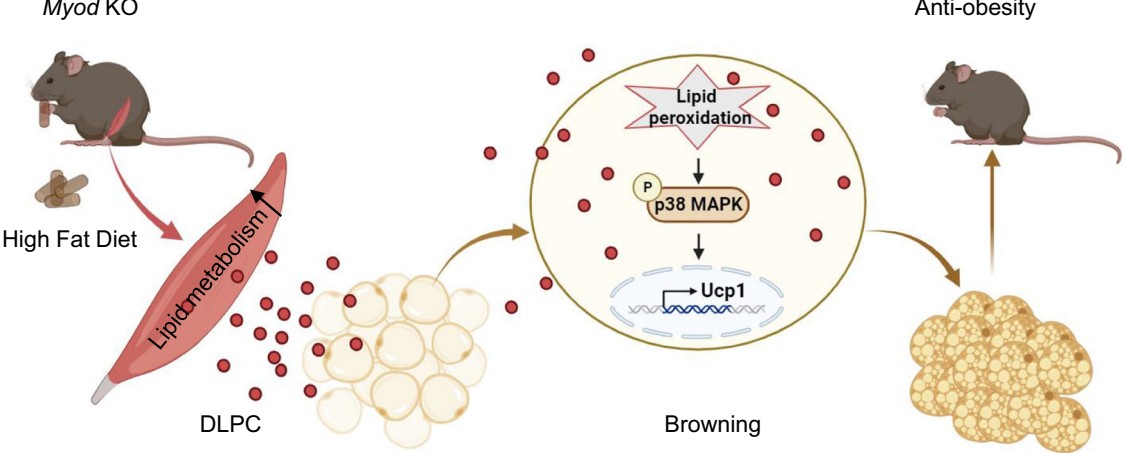

**Fig. 8 | Schematic for skeletal muscle-secreted DLPC inducing iWAT browning via lipid peroxidation-mediated p38 activation and orchestrating systemic energy homeostasis.** The *Myod* KO mice exhibit the elevated lipid metabolism in skeletal muscle tissues. Thus, there is increased levels of DLPC secreted from skeletal muscle of *Myod* KO mice. The skeletal muscle-derived DLPC induces lipid

peroxidation and activates p38 phosphorylation in white adipose tissues, thereby induces *Ucp1* gene expression and white adipose browning. The DLPC-induced iWAT browning benefits systemic energy homeostasis and contributes to the anti-obesity phenotype in *Myod* KO mice. Graphs in Fig. 8 is generated by BioRender (https://www.biorender.com).

L-Asparagine-$^{13}C_4$. Peak areas of endogenous metabolites were normalized to the areas of their corresponding isotopically labeled structural analogues for quantitation. For endogenous metabolites without labeled structural analogues, an automated algorithm selects the optimal internal standard for quantitation based on the rule of minimal coefficients of variations (COVs) after normalization[65].

### Protein extraction and peptide preparation

Mature adipocytes were lysed using sodium deoxycholate (SDC) buffer (4% SDC, 1% protease inhibitor cocktail, 100 mM Tris-Cl, pH 8.0), and the total protein content was determined using a BCA assay. Each phosphoproteomic sample was prepared according to a previous report[66]. In brief, 200 μg of proteins were reduced and alkylated with 10 mM Tris (2-carboxyethyl) phosphine hydrochloride (TCEP) and 40 mM 2-chloroacetamide (CAA) at 45 °C for 5 min, and then digested overnight with trypsin. Phosphorylated peptides were enriched by passing the sample through TiO₂ beads, and the TiO₂ beads were loaded into a in-house-generated C8 StageTip. The phosphorylated peptides were eluted from the TiO₂ beads using 5% ammonia/40% acetonitrile/55% water (v/v/v), dried, dissolved in 1% trifluoroacetic acid/99% isopropanol, and loaded to a homemade SDB-RPS StageTip for desalting. Each proteomic sample was prepared as described for phosphoproteomic sample preparation, but without the TiO₂ enrichment steps. After trypsin digestion, the tryptic peptides were desalted with a homemade SDB-RPS StageTip and used directly for MS analysis.

Redox proteomic sample preparation was performed according to the previously published activity-based protein profiling (ABPP) protocol with some modifications[67]. In brief, 1 mg of proteins were labeled with 100 μM iodoacetamide alkyne in 500 μL lysis buffer at 25 °C for 60 min, and then further incubated with 10 μL click chemistry reaction cocktail (500 mM NaVc, 500 mM THPTA, 100 mM CuSO₄, and 40 mM diazo biotin-azide). The reaction was incubated with vigorous shaking at 25 °C for 2 h. Labeled proteins were precipitated with prechilled acetone (−20 °C), dissolved in 500 μL 100 mM Tris-Cl, and digested overnight with 20 μg trypsin at 37 °C. The biotin-derivatized peptides were enriched using 20 μL of avidin beads and eluted with 100 mM Na₂S₂O₄. The eluted peptides were desalted on a homemade C18 StageTip and applied for MS analysis.

### LC-MS/MS analysis

Digested peptides were analyzed by liquid chromatography-tandem mass spectrometry (LC-MS/MS) using an Easy-nLC 1200 connected online to an Orbitrap Fusion Lumos mass spectrometer. A 250-mm Acclaim PepMap100 C18 column with an internal diameter of 75 μm was used to separate the peptides. Mobile phase A (0.1% FA in water) and mobile phase B (0.1% FA in ACN) were applied with a 60-min gradient as follows: 4 to 8% B in 2 min, 8 to 28% B in 43 min, 28 to 36% B in 8 min, 36 to 100% B in 2 min, and B at 100% for 5 min. The flow rate was set as 300 nL min-1.

The Orbitrap Fusion Lumos mass spectrometer was operated in the data-dependent acquisition mode. The FAIMS voltage was set as −40 v, −60 v, and −80 v. MS1 data were collected at a high resolution of 120,000 (m/z 200) with a mass range of 350-1500 m/z, a target value of 4e5, and a maximum injection time of 50 ms. For each full MS scan, the 20 most abundant precursor ions were selected for MS2 with an isolation window of 1.6 m/z and the HCD set with a normalized collision energy of 30. MS2 spectra were collected at a resolution of 15,000 (m/z 200). The target value was 5e4, the maximum fill time was 30 ms, and the dynamic exclusion time was 60 s.

### Proteomic, phosphoproteomic, and redox proteomic data analysis

The MS data were processed by MaxQuant (version 1.6.17.0), with Andromeda used as the database search engine[68]. Data were searched against the UniProt *Mus musculus* database (version 2021-03-12, 87975 sequences). Trypsin was chosen as the enzyme, with a maximum of two missed cleavages. For proteomic analysis, carbamidomethylation on cysteine (57.0214 Da) was considered a static modification, and oxidation on methionine (15.9949 Da) and protein N-terminal acetylation (42.0105 Da) were selected as dynamic modifications. For phosphoproteomic analysis, carbamidomethylation on cysteine (57.0214 Da) was considered a static modification, and oxidation on methionine (15.9949 Da), phosphorylation on serine/threonine/tyrosine (79.9663 Da), and protein N-terminal acetylation (42.0105 Da) were selected as dynamic modifications. For redox proteomic analysis, oxidation on methionine (15.9949 Da), carbamidomethylation on cysteine (57.0214 Da), and ABPP site on cysteine (273.1225 Da) were selected as dynamic modifications.

The phosphoproteomic and redox proteomic data was normalized according to protein abundance. The output protein list was analyzed and visualized using the DEP package, as described previously[69].

### RNA sequencing

Total RNA was isolated from TA and Sol muscles of mice fed with SD or HFD for 2 weeks, using the TRIzol reagent (Invitrogen). Sequencing libraries were generated using an NEBNext UltraTM RNA Library Prep Kit for Illumina (#E7530L, NEB) following the manufacturer's recommendations. Index codes were added to attribute sequences to each sample. Raw sequencing data were mapped to the mouse genome mm10 assembly using HISAT with default parameters. DEGSeq2 (Version 1.24.0) was used to calculate the read coverage for each gene. Differentially expressed genes (DEG) were filtered using criteria of >1.2-fold change and adjusted p-value < 0.05. The enrichment analysis of Gene Ontology (GO) terms among the differentially expressed genes was performed using ClusterProfiler (Version 3.14). A significance cutoff of $p$.adjust <0.05 and $q$ value < 0.05 was applied as the criteria for determining enriched GO terms.

### Real-time quantitative RT-PCR analyses

Total RNA was extracted from cells with the TRIzol reagent (Invitrogen) and reverse-transcribed (RT) using RevertAid reverse transcriptase (Thermo Scientific, EP0442). For measuring mature mRNAs, quantitative PCR (qPCR) analyses were performed with the SosoFast qPCR Master Mix (Bio-Rad, 1725202) using an iQ5 Multicolor Real-Time PCR Detection System (Bio-Rad). PPIA served as an internal control. All primers used for RT-qPCR were included in Source data.

### Western blot analysis

Mouse tissues or cells were lysed in RIPA lysis buffer containing protease and phosphatase inhibitors. Protein lysates were resolved by SDS-PAGE, transferred to a polyvinylidene fluoride (PVDF) membrane, and immunoblotted with primary antibodies against total p38 (1:1000, Cell Signaling Technology, 9212 S, Rabbit polyclonal), phosphorylated

p38 (T180/Y182, 1:1000, Cell Signaling Technology, 9211 S, Rabbit polyclonal), Ucp1 (1:300, Abcam, ab10983, Rabbit polyclonal), β-tubulin (1:4000, Engibody Biotechnology, AT0003, Mouse monoclonal, clone number: 6C4). Membranes were washed in TBS washing buffer for 30 min, incubated with horseradish peroxidase (HRP)−conjugated secondary antibodies (Zhongshanjinqiao Corporation, 1:2000) for 1 h at room temperature, and washed in TBS washing buffer for 30 min. Each membrane was then placed into ECL detection solution (Thermo Scientific 34578), incubated for 1 min at room temperature, and subsequently exposed to a chemiluminescence instrument (Tanon #5800). Full scans of all Western blots are available in Source Data.

### Immunofluorescent staining

Cells were fixed with 4% paraformaldehyde in PBS at room temperature for 15 min. After the fixation, cells were incubated with 0.1% Triton X-100 in PBS at room temperature for 10 min and further incubated for 1 h in blocking buffer (3% BSA). Subsequently, immunostaining was performed according to standard protocols using primary antibodies against MHC (1:500, DSHB, MF20, Mouse monoclonal). Primary antibodies diluted with PBS buffer containing 3% BSA were applied, and the samples were incubated overnight at 4°C. Cells were then washed with PBS and incubated for 1 h with Alexa Fluor 647 fluorescein-conjugated secondary antibodies (1:1000, Invitrogen, A32787). After three times washes with PBS, the samples were imaged under a fluorescence microscope (Inverted Microscope Leica DMi8).

### Histological analysis

Tissues were fixed overnight in 4% (v/v) PFA and embedded in paraffin wax. The embedded samples were cut and mounted on glass slides. The slide-mounted samples were stained with hematoxylin and eosin (H&E).

### Statistical analysis

Statistical analysis was performed with the GraphPad Prism (Version 9.4.1) software. For statistical comparisons of two conditions, the two-tail Student's $t$-test was used. Comparisons of multiple groups were made using one- or two-way ANOVA. Data are presented as the mean ± SD. The following applies for all figures: $*p < 0.05$, $**p < 0.01$, $***p < 0.001$, and $****p < 0.0001$.

### Reporting summary

Further information on research design is available in the Nature Portfolio Reporting Summary linked to this article.

## Data availability

The raw sequence data reported in this paper have been deposited in the Genome Sequence Archive[70] in National Genomics Data Center[71], China National Center for Bioinformation / Beijing Institute of Genomics, Chinese Academy of Sciences (GSA: CRA010866) that are publicly accessible at https://ngdc.cncb.ac.cn/search/?dbId=gsa&q=CRA010866. All mass spectrometry proteomics raw data have been deposited to the ProteomeXchange Consortium (http://proteomecentral.proteomexchange.org) via the iProX partner repository (https://www.iprox.cn//page/SCV017.html?query=%20IPX0007379000) with the dataset identifier PXD046337. Source data are provided with this manuscript.

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

## Acknowledgements

This work was supported by grants from the National Key R&D Program of China (2022YFA0806002, 2021YFA1100202), the National Natural Science Foundation of China (31971080, 91949106 and 32000603), the Basic Research Projects of Basic Strengthening Program (2020-JCJQ-ZD-264), and the CAMS Innovation Fund for Medical Sciences (2021-I2M-1-016).

## Author contributions

Conceptualization, D.Z. and Y.Z.; Investigation, X.H., M.S., Q.C., YX.Z. and N.L.; Formal analysis, S.W., P.Y., Y.Y. and S.M.L.; Methodology, Q.Z., A.T., Y.G., X.W., M.C., H.L., X.Z., G.S., S.F., L.Z., P.T. and C.C.L.; Writing – Original Draft, Y.Z. and X.H.; Writing – Review & Editing, D.Z., Y.Z. and S.F.; Funding Acquisition, D.Z., Y.Z., M.C. and H.L.; Supervision, D.Z., Y.Z. and H.L.

## Competing interests

The authors declare no competing interests.
