## [Peer Review File · Nature Communications]

REVIEWER COMMENTS

Reviewer #1 (Remarks to the Author):

This study identifies DLPC as a muscle derived lipid (myolipokine) that plays a role in regulating systemic metabolism through promoting browning of adipose tissues. The result is surprising, exciting and novel. The analysis is quite extensive, including multiomics analyses of high diet treated and knockout mice. In general, the experiments are well designed and the conclusion on the effect of DLPC on preventing and ameliorating HFD-induced obesity is solid. However, the part linking DLPC to muscle and Myod1 KO is not particular strong. Specifically, the authors initially emphasized how Myod1 KO elevated oxidative metabolism in the muscle, which should improve systemic metabolism especially fatty acid metabolism. Unfortunately, the rest of the study hinges on adipose browning as a single factor attributing to the overall metabolic phenotype, without considering the contribution of skeletal muscles to the overall (systemic) metabolic phenotype. In addition, there is insufficient evidence to prove that the source of DLPC in the circulation is from muscle cells, and therefore be classified as a myolipokine. As this is one of the most novel parts of the study, some validation should be considered to support that DLPC is indeed from the muscle cells.

1. Relative contribution of the skeletal muscle to the altered systemic energy metabolism should be assessed. A contribution (at least partial contribution) from the muscle is expected as the authors show that Myod KO elevated oxidative metabolism and fatty acid oxidation in the muscle. This should also improve resistance to HFD-induced obesity. To what extent the muscle and adipose brown contribute to the overall systemic metabolism should be dissected.
2. The results demonstrating muscle-secretion of DLPC is performed on whole muscles, which contains many non-muscle cells such as FAPs. Although unlikely, it is possible that the non-muscle cells may have contributed to the elevated level of PC 36:4 in the whole muscle conditioned medium and Myod^{-/-} serum. Direct demonstration that DLPC is secreted by skeletal muscle cells would strengthen the conclusion that DLPC is a myolipokine. As the authors have done RNA-seq, are genes and pathways involved in DLPC (or PC) biosynthesis altered by Myod KO? Alternatively, myofibers (instead of whole muscles) from WT and KO mice could be analyzed for DLPC content.
3. Demonstrating the effect of DLPC supplementation on adipose tissue is nice. As control, does DLPC have similar effects in muscle cells on p38 phosphorylation? There is evidence that p38 MAPK alters muscle metabolism.

Minor:

1. Previous studies have reported a role of Myod in controlling Prdm16 and adipocyte browning through miR-133 and that Myod KO promotes transdifferentiation of myoblasts into brown adipocytes. These results are relevant to the current finding and should be discussed in the manuscript.
2. GTT and ITT results should include area under curve (AUC) and run statistical analysis on the significance.
3. Line 153-154: “elevated oxidative metabolism in Myod KO muscle might contribute to differences in whole-body metabolic homeostasis”. First, please provide evidence or cite references to show elevated oxidative metabolism in KO muscles. In this study, the authors provided only evidence for related changes in gene expression but not functional evidence. Second, changes in muscle metabolism should be taken into account in later analysis of systemic metabolism (related to major comments #1).
4. Line 193-197: Gene expression analysis indicates no changes in lipolysis related genes, and the authors then conclude that lipolysis does not contribute to the phenotype. This statement is problematic. In addition to protein levels, hormonal activation of ATGL, HSL also plays a role in regulating lipolysis. This statement also contradicts later results showing browning – shouldn't browning itself promotes lipolysis or requires elevated lipolysis?

Reviewer #2 (Remarks to the Author):

This work aimed at uncovering a putative muscle-secreted lipokine, PC(18:2/18:2), and its function in maintaining metabolic homeostasis. It is suggested that PC(18:2/18:2) induces adipose browning via lipid peroxidation-mediated p38. Throughout the manuscript, the generated data also helped uncover the molecular consequences of the genetic deletion of Myod in mice. Overall, this is an exciting finding with the potential for subsequent development of an efficient treatment for obesity in humans. I have a few comments to hopefully make it a better manuscript. Please refer to the below comments for more information.

Main concerns and comments:

- In the transcriptomics analysis, the raw p-value of 0.05 was used (together with the FC) as the significant cut-off. Using raw significance cut-off is inappropriate in the context of high-dimensional data analysis. I would instead suggest a significant cut-off of the raw p-value of 0.05 and an adjusted p-value (e.g., False-Discovery Rate) of up to 0.25 in the context of the exploratory research. This should be considered and revised accordingly. This is particularly crucial as the adjusted p-value was used in some of the other analyses.

- In lines 127-136, it is unclear whether there was a head-to-head statistical analysis from the raw data (e.g., HFD(sol) vs HFD(TA)) or it was some sort of Venn's diagram-based interpretation. The latter approach is statistically weak and might be misleading.
- In lines 139-140, it is stated that the expression of metabolic enzyme-encoding genes was explored. However, the source of definition of the included genes is unclear, i.e., based on which reference materials, knowledgebase, or database. As far as I am concerned, there is not yet a consensus for defining a group of genes/proteins among available sources.
- The section called "iWAT browning occurs in Myod KO mice" is said to be a section about "molecular mechanism". However, the observations, in my humble opinion, are instead the level of cellular phenotypic observations to specify the molecular processes involved.
- Throughout the manuscript, different lengths of the treatment period (experimentation) were used (e.g., 8 weeks, 10 weeks, 12 weeks). Perhaps there was some sort of additional biochemical data to guide the selection? Please clarify this point.
- This point may seem trivial, but to me, it is significant. Data must be presented as mean and standard deviation (SD) rather than the standard error of the mean (SEM). SEM is a measure of the precision of the estimated population mean. It does not present the data variability around mean, hence not a descriptive statistic at all. The use of SEM is preferred simply because it makes the 'plots' look better, but biologically irrelevant.
- It is better to summarize the whole research findings via a schematic figure. It would help the reader get a precise idea about this sophisticated work much faster.
- In Figure 4C, how was the lipid-class level summarization conducted and standardized? The data handling, transformation and standardization process should be described as this data type could be wrongly handled in the context of LC-MS data.
- One of the significant findings in this work is the energy homeostasis role of a PC with two acyl chains of two levels of unsaturation each. Any thoughts on and discussion possible related to polyunsaturated fatty acyl lipids in general? Besides, since soybean PC did contain a few PC species, how do the authors think about the risk of missing potentially significant PCs?
- Please state the limitations of the current study to pave ways for subsequent investigations.

Minor comments:

- The opening sentences of the abstract lack a 'hook' to help the reader realize the importance of this work. Also, in the abstract, the HFD was not defined. Please revise the abstract accordingly.
- In the introduction, lines 54-56, the sentence could be a bit more specific.
- It is not clear why in line 65, it is written that "many metabolites and intermediate metabolites", are there any specific reasons to differentiate the two terms? Please clarify if "intermediate metabolites" should be emphasized.
- There is a lack of references for the provided information in lines 69-72.

- It is better to define the word iWAT (line 98). On the other hand, the HFD definition was defined again in line 110, which should be removed.

- In line 135, it is written that “might be unfavorable for the ability”, isn't it better to use an alternative word, e.g., “not efficient”?

- In lines 144-149, should the gene nomenclature be checked to be consistent for the mice model? A similar issue happens occasionally across the manuscript (or it is just due to my lack of understanding).

- In line 167, should the “protect” be changed to “protective effects” or something similar?

- In lines 183, the “decreased levels” used in the sentence seems odd. Perhaps, “lower level” or something similar fits the sentence better.

- In lines 240-241, it is not appropriate to write “PC could be metabolite that”. Here, PC indicates a subclass of lipids.

Point-by-point response for manuscript (NCOMMS-23-23713-T)

Reviewer #1:

This study identifies DLPC as a muscle derived lipid (myolipokine) that plays a role in regulating systemic metabolism through promoting browning of adipose tissues. The result is surprising, exciting and novel. The analysis is quite extensive, including multiomics analyses of high diet treated and knockout mice. In general, the experiments are well designed and the conclusion on the effect of DLPC on preventing and ameliorating HFD-induced obesity is solid. However, the part linking DLPC to muscle and Myod1 KO is not particular strong. Specifically, the authors initially emphasized how Myod1 KO elevated oxidative metabolism in the muscle, which should improve systemic metabolism especially fatty acid metabolism. Unfortunately, the rest of the study hinges on adipose browning as a single factor attributing to the overall metabolic phenotype, without considering the contribution of skeletal muscles to the overall (systemic) metabolic phenotype. In addition, there is insufficient evidence to prove that the source of DLPC in the circulation is from muscle cells, and therefore be classified as a myolipokine. As this is one of the most novel parts of the study, some validation should be considered to support that DLPC is indeed from the muscle cells.

1. *Relative contribution of the skeletal muscle to the altered systemic energy metabolism should be assessed. A contribution (at least partial contribution) from the muscle is expected as the authors show that Myod KO elevated oxidative metabolism and fatty acid oxidation in the muscle. This should also improve resistance to HFD-induced obesity. To what extent the muscle and adipose brown contribute to the overall systemic metabolism should be dissected.*

Reply: Thank you for your constructive suggestions. We assessed relative contribution of the skeletal muscle to the altered systemic energy metabolism in *Myod* KO mice. As both muscle heat production and white adipose browning significantly contribute to keep body temperature and metabolic homeostasis, we firstly measured core temperature of *Myod* KO and their WT controls. The core temperature in *Myod* KO mice were significantly higher than that in WT controls at either thermal neutral or cold conditions (**Reviewer Fig. 1a and 1b**), which is consistent with our previous metabolic chamber analysis showing higher heat production or energy expenditure in *Myod* KO mice than in WT controls.

To further evaluate muscle contribution for the higher heat production/energy expenditure in *Myod* KO mice, we performed mitochondrial respiration assay in the differentiated myotubes by Seahorse XFe24. To this end, the primary myoblasts isolated from the *Myod*^{fl/fl} mice were induced to differentiation for 48 h and then infected with adenovirus expressing Cre (Ad-Cre) to achieve deletion of *Myod* (cKO), infection with the adenovirus expressing EGFP (Ad-EGFP) as controls (cWT) (**Reviewer Fig. 1c**). This approach achieved successful knockout of *Myod* in the cKO myotubes, as measured by RT-qPCR (**Reviewer Fig. 1d**) and the cKO myotubes were as healthy as cWT controls (**Reviewer Fig. 1e**). Thus, the mitochondrial respiration assay was performed with the myotubes by Seahorse XFe24. Consistently, we found that the cKO myotubes exhibited higher oxygen consumption rate (OCR) than that in cWT controls (**Reviewer Fig. 1f**). Although, our data suggest that skeletal muscle might have a partial contribution for resistance to HFD-induced obesity in the *Myod* KO mice, this intriguing finding is quite preliminary and we are currently performing

more experiments to validate this observation and are dedicated to accumulating more robust evidence to shed light on this matter in our future research endeavors.

Reviewer Fig.1 Knockout of *Myod* might enhance mitochondrial respiration.

(a) Core temperature in *Myod* KO mice (KO) ($n = 13$) and their WT controls ($n = 12$) at the thermal neutral conditions.

(b) Core temperature in KO mice ($n = 13$) and their WT controls ($n = 12$) at 4°C for the indicated

time points (0, 1, 2, 3, 4 h).

(c) Schematic diagram showing acute deletion of *Myod* in differentiated myotubes. The primary myoblasts isolated from the *Myod*^{fl/fl} mice were induced to differentiation for 48 h and then infected with adenovirus expressing Cre (Ad-Cre) to achieve deletion of *Myod* (cKO), infection with the adenovirus expressing EGFP (Ad-EGFP) as controls (cWT).

(d) Relative levels of *Myod* mRNA in cKO and cWT myotubes, as determined by RT-qPCR.

(e) Representative images showing immunostaining of myosin heavy chain (MHC) (red), a marker for the differentiated myotube, in cKO and cWT myotubes. DAPI (blue) served to visualize nuclei. Scale bar, 100 μ m.

(f) Oxygen consumption rate (OCR), determined by Seahorse XFe24.

Data are presented as mean \pm SD. Significance was assessed by two-way ANOVA (b, f) or two tail Student's *t*-test (a, d). **p* < 0.05, ***p* < 0.01, ****p* < 0.001, *****p* < 0.0001 compared to WT or cWT control group.

2. The results demonstrating muscle-secretion of DLPC is performed on whole muscles, which contains many non-muscle cells such as FAPs. Although unlikely, it is possible that the non-muscle cells may have contributed to the elevated level of PC 36:4 in the whole muscle conditioned medium and Myod^{-/-} serum. Direct demonstration that DLPC is secreted by skeletal muscle cells would strengthen the conclusion that DLPC is a myolipokine. As the authors have done RNA-seq, are genes and pathways involved in DLPC (or PC) biosynthesis altered by Myod KO? Alternatively, myofibers (instead of whole muscles) from WT and KO mice could be analyzed for DLPC content.

Reply: Thank you for the constructive comments. As suggested, we directly validated that DLPC is a bona fide myolipokine secreted by skeletal muscle per se. To this end, we isolated primary myoblasts and induced the cells for differentiation to myotubes. The resulting myotubes were further cultured in serum-free DMEM for 24 hours, and then the conditioned medium was collected for lipidomics analysis (**Reviewer Fig. 2a**). Compared to DMEM control, we indeed detected DLPC in the conditioned medium (CM) (**Reviewer Fig. 2b**). Thus, our data convincingly support the notion that DLPC is a muscle-secreted lipokine (myolipokine).

As you suggested, we also analyzed our RNA-seq data from *Soleus* muscle of *Myod* KO and WT littermates fed with high fat diet (HFD) or standard diet (SD). The data showed the increased mRNA levels of the genes involved in the PC biosynthesis pathway in *Myod* KO mice compared to WT controls in either SD or HFD conditions (**Reviewer Fig. 3**), indicating that skeletal muscle of *Myod* KO mice produce more PC than WT littermates. Together, we provided biochemical and molecular evidence to support our conclusion that DLPC is a bona fide muscle-secreted lipokine (myolipokine).

Reviewer Fig. 2 DLPC is detected in the conditioned medium from the differentiated myotubes.

(a) Experimental scheme for using lipidomics analyses to detect myotubes-secreted DLPC from conditioned medium (CM).

(b) Levels of DLPC (18:2/ 18:2) in myotubes-derived CM.

Data are presented as mean ± SD. Significance was assessed by two tail Student's *t*-test (b). ***p* < 0.01, compared to DMEM control.

Reviewer Fig. 3 Expression levels of the genes involved in the PC biosynthesis pathway in *Soleus* muscle of *Myod* KO mice compared to WT controls fed with SD or HFD, respectively.

(a-l) The data (TPM) were determined by RNA-seq.

Data are presented as mean \pm SD. Significance was assessed by two tail Student's *t*-test (a-l). * $p < 0.05$, compared to WT, $n = 3$ for each group.

3. *Demonstrating the effect of DLPC supplementation on adipose tissue is nice. As control, does DLPC have similar effects in muscle cells on p38 phosphorylation? There is evidence that p38 MAPK alters muscle metabolism.*

Reply: As suggested, we examined p38 phosphorylation in TA muscle of the mice i.p administered with 200 mg/kg dose of DLPC for 30 min. We found that DLPC activated p38 phosphorylation in skeletal muscle (**Reviewer Fig. 4**). This observation is consistent with previously published data that p38 MAPK signaling regulates oxidative metabolism in skeletal muscle¹⁻⁵. However, the functional role and underlying mechanism(s) of MyoD-mediated p38 activation in skeletal muscle of MyoD KO mice is completely unknown and is interesting to be investigated further.

References

1. Akimoto, T. *et al.* Exercise stimulates Pgc-1alpha transcription in skeletal muscle through activation of the p38 MAPK pathway. *J Biol Chem* **280**, 19587–19593 (2005).
2. Yoon, M.J. *et al.* Adiponectin increases fatty acid oxidation in skeletal muscle cells by sequential activation of AMP-activated protein kinase, p38 mitogen-activated protein kinase, and peroxisome proliferator-activated receptor alpha. *Diabetes* **55**, 2562–2570 (2006).
3. Pogozelski, A.R. *et al.* p38gamma mitogen-activated protein kinase is a key regulator in skeletal muscle metabolic adaptation in mice. *PLoS One* **4**, e7934 (2009).
4. Jeong, H.J. *et al.* Prmt7 Deficiency Causes Reduced Skeletal Muscle Oxidative Metabolism and Age-Related Obesity. *Diabetes* **65**, 1868–1882 (2016).
5. Bengal, E., Aviram, S. & Hayek, T. p38 MAPK in Glucose Metabolism of Skeletal Muscle: Beneficial or Harmful? *Int J Mol Sci* **21** (2020).

Reviewer Fig.4 Phosphorylation of p38 in TA muscle of the C57BL/6j mice treated with DLPC. Representative Western blots showing p38 phosphorylation in TA from C57BL/6j mice i.p. administered with vehicle or 200 mg/kg DLPC for 30 min. β -Tubulin served as an equal loading control, $n = 4$ for each group.

Minor:

1. *Previous studies have reported a role of Myod in controlling Prdm16 and adipocyte browning through miR-133 and that Myod KO promotes transdifferentiation of myoblasts into brown adipocytes. These results are relevant to the current finding and should be discussed in the manuscript.*

Reply: Thank you for your suggestions. We have discussed the relevant work (Wang et al, 2017) in the introduction part of the revised version of the manuscript.

Wang C, Liu W, Nie Y, Qaher M, Horton HE, Yue F, Asakura A, Kuang S. **Loss of MyoD Promotes Fate Transdifferentiation of Myoblasts Into Brown Adipocytes.** EBioMedicine. 2017 Feb;16:212-223.

2. *GTT and ITT results should include area under curve (AUC) and run statistical analysis on the significance.*

Reply: As you suggested, we have calculated the area under curve (AUC) and performed statistical analysis. The data have been included in the revised version of the manuscript.

3. *Line 153-154: “elevated oxidative metabolism in Myod KO muscle might contribute to differences in whole-body metabolic homeostasis”. First, please provide evidence or cite references to show elevated oxidative metabolism in KO muscles. In this study, the authors provided only evidence for related changes in gene expression but not functional evidence. Second, changes in muscle metabolism should be taken into account in later analysis of systemic metabolism (related to major comments #1).*

Reply: Thank you for your constructive suggestions and comments. To show elevated oxidative metabolism in KO muscles, we performed mitochondrial respiration assay in the differentiated myotubes by Seahorse XFe24 (**Reviewer Fig. 1c and 1f**). Please refer to Question #1 raised by this reviewer; we have discussed the enhanced oxidative metabolism in muscle contributes to systemic homeostasis in the revised version of the manuscript.

4. *Line 193-197: Gene expression analysis indicates no changes in lipolysis related genes, and the authors then conclude that lipolysis does not contribute to the phenotype. This statement is problematic. In addition to protein levels, hormonal activation of ATGL, HSL also plays a role in regulating lipolysis. This statement also contradicts later results showing browning – shouldn't browning itself promotes lipolysis or requires elevated lipolysis?*

Reply: Thank you for your suggestions and comments. We have rephrased the sentence in the revised version of the manuscript.

Reviewer #2:

This work aimed at uncovering a putative muscle-secreted lipokine, PC (18:2/18:2), and its function in maintaining metabolic homeostasis. It is suggested that PC (18:2/18:2) induces adipose browning via lipid peroxidation-mediated p38. Throughout the manuscript, the generated data also helped uncover the molecular consequences of the genetic deletion of Myod in mice. Overall, this is an exciting finding with the potential for subsequent development of an efficient treatment for obesity in humans. I have a few comments to hopefully make it a better manuscript. Please refer to the below comments for more information.

Main concerns and comments:

- *In the transcriptomics analysis, the raw p-value of 0.05 was used (together with the FC) as the significant cut-off. Using raw significance cut-off is inappropriate in the context of high-dimensional data analysis. I would instead suggest a significant cut-off of the raw p-value of 0.05 and an adjusted p-value (e.g., False-Discovery Rate) of up to 0.25 in the context of the exploratory research. This should be considered and revised accordingly. This is particularly crucial as the adjusted p-value was used in some of the other analyses.*

Reply: Thank you for your constructive suggestions and comments. Actually, we indeed used an adjusted p-value but not p-value for the indicated transcriptomics analysis. We apologized for our typo mistake in manuscript preparation and have corrected this in the revised version of the manuscript.

- *In lines 127-136, it is unclear whether there was a head-to-head statistical analysis from the raw data (e.g., HFD(sol) vs HFD(TA)) or it was some sort of Venn's diagram-based interpretation. The latter approach is statistically weak and might be misleading.*

Reply: Thank you for your comments. Yes. We performed a head-to-head statistical analysis from the raw data, including SD-TA vs HFD-TA, SD-Sol vs HFD-Sol, to identify the differentially expressed TFs between TA and Sol muscle in response to HFD.

- *In lines 139-140, it is stated that the expression of metabolic enzyme-encoding genes was explored. However, the source of definition of the included genes is unclear, i.e., based on which reference materials, knowledgebase, or database. As far as I am concerned, there is not yet a consensus for defining a group of genes/proteins among available sources.*

Reply: Thank you for your suggestions. Yes. The metabolic enzyme-encoding genes examined in the study were previously reported in the literature. We have cited the references in the revised version of the manuscript.

- *The section called "iWAT browning occurs in Myod KO mice" is said to be a section about "molecular mechanism". However, the observations, in my humble opinion, are instead the level of cellular phenotypic observations to specify the molecular processes involved.*

Reply: Thank you for this nice comment. According, we have rephrased the first sentence of the section in the revised version of the manuscript.

- *Throughout the manuscript, different lengths of the treatment period (experimentation) were used (e.g., 8 weeks, 10 weeks, 12 weeks). Perhaps there was some sort of additional biochemical data to guide the selection? Please clarify this point.*

Reply: Thank you for pointing this out. We apologized for making such confused description. We have corrected the description in the revised version of the manuscript.

- (1) For the metabolic phenotype observation, the *Myod* KO and WT mice were fed with HFD or SD for 12 weeks.
- (2) For DLPC prevention experiment, the C57BL/6j mice were fed with HFD and intraperitoneally administrated with DLPC for 14 weeks.
- (3) For DLPC treatment experiment, the C57BL/6j mice were fed with HFD or SD for 8 weeks

followed by DLPC-treated for 10 weeks.

- This point may seem trivial, but to me, it is significant. Data must be presented as mean and standard deviation (SD) rather than the standard error of the mean (SEM). SEM is a measure of the precision of the estimated population mean. It does not present the data variability around mean, hence not a descriptive statistic at all. The use of SEM is preferred simply because it makes the 'plots' look better, but biologically irrelevant.

Reply: Thank you very much for your positive comments. We have reanalyzed the data and presented as mean \pm SD in the revised version of the manuscript.

- It is better to summarize the whole research findings via a schematic figure. It would help the reader get a precise idea about this sophisticated work much faster.

Reply: Thank you for your constructive suggestions. We have prepared a schematic figure and placed in the revised version of the manuscript.

- In Figure 4C, how was the lipid-class level summarization conducted and standardized? The data handling, transformation and standardization process should be described as this data type could be wrongly handled in the context of LC-MS data.

Reply: Thank you for your suggestions and comments. The heatmaps from lipidomics analysis were generated using pheatmap package in R environment, and the data were z-score normalized by rows.

- One of the significant findings in this work is the energy homeostasis role of a PC with two acyl chains of two levels of unsaturation each. Any thoughts on and discussion possible related to polyunsaturated fatty acyl lipids in general? Besides, since soybean PC did contain a few PC species, how do the authors think about the risk of missing potentially significant PCs?

Reply: Thank you for your concerns. We searched literature and found no polyunsaturated fatty acyl lipids to be documented for regulating whole-body energy homeostasis. As soybean PCs contain a few PC species, we could not rule out the possibility that additional types of PC might regulate whole body energy homeostasis and we have discussed this point in the revised version of the manuscript.

- Please state the limitations of the current study to pave ways for subsequent investigations.

Reply: Thank you for your suggestions. We have added a statement for the limitations of the current study in the revised version of the manuscript.

Minor comments:

- The opening sentences of the abstract lack a 'hook' to help the reader realize the importance of this work. Also, in the abstract, the HFD was not defined. Please revise the abstract accordingly.

Reply: As suggested, we have revised the abstract accordingly.

- In the introduction, lines 54-56, the sentence could be a bit more specific.

Reply: Thank you for your suggestions. We have rephrased the sentence in the revised version of the manuscript.

- *It is not clear why in line 65, it is written that “many metabolites and intermediate metabolites”, are there any specific reasons to differentiate the two terms? Please clarify if “intermediate metabolites” should be emphasized.*

Reply: Thank you for your suggestions. We have rephrased the sentence in the revised version of the manuscript.

- *There is a lack of references for the provided information in lines 69-72.*

Reply: Thank you very much. We have added the related reference in the revised version of the manuscript.

- *It is better to define the word iWAT (line 98). On the other hand, the HFD definition was defined again in line 110, which should be removed.*

Reply: Thank you for pointing this. We have corrected these in the revised version of the manuscript.

- *In line 135, it is written that “might be unfavorable for the ability”, isn’t it better to use an alternative word, e.g., “not efficient”?*

Reply: Thanks for the nice suggestion and this sentence has been corrected in the revised version of the manuscript.

- *In lines 144-149, should the gene nomenclature be checked to be consistent for the mice model? A similar issue happens occasionally across the manuscript (or it is just due to my lack of understanding).*

Reply: Thank you for your suggestions. We have checked the gene nomenclature to be consistent throughout the manuscript.

- *In line 167, should the “protect” be changed to “protective effects” or something similar?*

Reply: As suggested, we have changed “protect” to “protective effects” in the revised version of the manuscript.

- *In lines 183, the “decreased levels” used in the sentence seems odd. Perhaps, “lower level” or something similar fits the sentence better.*

Reply: Thanks very much and we have corrected the sentence in the revised version of the manuscript.

- *In lines 240-241, it is not appropriate to write “PC could be metabolite that”. Here, PC indicates a sub-class of lipids.*

Reply: Thank you for your suggestions. The sentence is rephrased in the revised version of the manuscript.

REVIEWERS' COMMENTS

Reviewer #1 (Remarks to the Author):

The authors did a nice job addressing my previous comments. I have no further major comments other than a couple of editorial issues

Minor:

The last sentence of the first paragraph in introduction about lipokine is inaccurate.

"Interestingly, recent studies have demonstrated that lipokines, which are lipid species predominantly secreted from adipose tissue, play critical roles in controlling whole-body energy homeostasis by communicating with other metabolic organs, such as liver and muscle". I believe that lipokines also include proteins secreted from adipocytes.

Unless the author responses will published along with the online version, it would have been nice to include the figures in the author response as formal or supplementary figures in the manuscripts.

Reviewer #2 (Remarks to the Author):

The manuscript has been revised carefully. It may be accepted for publication.

Point-by-point response for manuscript (NCOMMS-23-23713A)

Reviewer #1:

The authors did a nice job addressing my previous comments. I have no further major comments other than a couple of editorial issues

Minor:

The last sentence of the first paragraph in introduction about lipokine is inaccurate.

"Interestingly, recent studies have demonstrated that lipokines, which are lipid species predominantly secreted from adipose tissue, play critical roles in controlling whole-body energy homeostasis by communicating with other metabolic organs, such as liver and muscle". I believe that lipokines also include proteins secreted from adipocytes.

Reply: Thank you for your concern. Adipose-derived signalling molecules, including adipokines and lipokines, represent a novel regulatory network in maintaining systemic homeostasis. The bioactive peptides or proteins secreted from adipocytes are generally termed as adipokines^{1,2}. The lipid species derived from adipose tissue termed as lipokines, including lysophosphatidic acid (LPA), palmitoleate (C16:1n7), fatty acid hydroxy fatty acids (FAHFAs), and oxylipin^{2,3}. Therefore, we used the term lipokines to indicate adipose-derived lipid species in controlling whole-body energy homeostasis.

1. Rodríguez A, Becerril S, Hernández-Pardos AW, Frühbeck G. Adipose tissue depot differences in adipokines and effects on skeletal and cardiac muscle. *Curr Opin Pharmacol*. 2020 Jun;52:1-8.
2. Gu X, Wang L, Liu S, Shan T. Adipose tissue adipokines and lipokines: Functions and regulatory mechanism in skeletal muscle development and homeostasis. *Metabolism*. 2023 Feb;139:155379.
3. Li VL, Kim JT, Long JZ. Adipose Tissue Lipokines: Recent Progress and Future Directions. *Diabetes*. 2020 Dec;69(12):2541-2548.

Unless the author responses will published along with the online version, it would have been nice to include the figures in the author response as formal or supplementary figures in the manuscripts.

Reply: Thank you for your suggestion. We have placed the reviewer figures in the supplementary figures and agree to publish our response along with the online version of the supplementary information.

Reviewer #2:

The manuscript has been revised carefully. It may be accepted for publication.